# Evaluating the Effectiveness of Nanotechnology in Environmental Remediation of a Highly Metal-Contaminated Area—Minas Gerais, Brazil

**Rita Fonseca** [1,2,3,*], **Joana Araújo** [2,3], **Catarina Pinho** [2,3] **and Teresa Albuquerque** [2,4]

1    Department of Geosciences, School of Sciences and Technologies, University of Évora, 7004-516 Evora, Portugal
2    Institute of Earth Sciences (IES), University of Évora Pole, 7004-516 Evora, Portugal; jfaraujo@uevora.pt (J.A.); c_pinho@uevora.pt (C.P.); teresal@ipcb.pt (T.A.)
3    AmbiTerra Laboratory, University of Évora, Colégio Pedro da Fonseca, 7005-345 Evora, Portugal
4    Polytechnic Institute of Castelo Branco/CERNAS, 6000-084 Castelo Branco, Portugal
*    Correspondence: rfonseca@uevora.pt

**Abstract:** A column experiment at a laboratory level was carried out to assess the effect of the application of nanotechnology in the decontamination of soils and alluvial deposits with high levels of potentially toxic elements (PTEs). A suspension of zero-valent iron nanoparticles (nZVI) was injected at three different concentrations in selected samples (two sediments, one soil). For most of the elements, the retention by nZVI was proportional to the concentration of the suspension and the trend was similar. Metals were immobilized by adsorption on the surface layer of the nanoparticles and/or by complexation, co-precipitation, and chemical reduction. By day 60 following injection, the nZVI lost reactivity and the retained species were desorbed and back into the soluble phase. The definition of spatial patterns for PTEs' distribution allowed for the construction of contamination risk maps using a geostatistical simulation approach. The analysis obtained from the extractable contents of five target elements (Zn, Cu, Cd, Pb, As) was cross-checked with the estimated map network to assess their retention efficiency. Data from the analysis of these elements, in the extractable phase and in the porewater of the sediments/soils, indicate the nZVI injection as a suitable technique for reducing the risk level of PTEs in contaminated Fe-rich tropical environments.

**Keywords:** metal contaminants; nanoparticles of zero-valent iron; laboratory scale experiment; metals adsorption; sequential gaussian simulation

## 1. Introduction

This work is part of a wider study on the evaluation of the risk level and of the effectiveness of the application of remediation technologies in a highly metal-contaminated area around an industrial unit located on the southern riverbank of the largest Brazilian river, San Francisco, in Minas Gerais State [1]. This is an important zinc treatment industrial plant that produces SHG Zinc, metallic alloys, and zinc oxides, from the metallurgical treatment of silicate (85%) and sulphate (15%) ores [2,3]. Since the late 1950s, this unit has a large volume of zinc processing, starting at about 10 thousand ton/y, which reached 90 thousand ton/y in the 70's, and near 200 thousand ton/y, nowadays [2,4,5]. This industrial treatment and production resulted in the generation of Zn-rich industrial wastes and their by-products, such as Cd, Zn, Pb, Cu, and As, which, until the 1970s, were stored as a surficial uncovered dump, mixed with local scrap dumps, construction wastes, oil drums, and waste of personal protective equipment, around the plant, near the banks of the São Francisco River, and near its tributaries. In the absence of appropriate regulations and legislation on waste management and disposal systems, the early activity of this industrial plant triggered several environmental problems, in surface waters, sediments, and soils of the surrounding area and in the São Francisco River. To avoid such negative effects

on the local environment and pressured by the responsible state environmental agencies, active efforts towards pollution reduction have been pursued by the metallurgical company. During the last 40 years, there was many interventions trying to contain water and solid contamination. Waste materials were placed in containment structures, which proved to be ineffective in a long term [6]. Since 2011, waste from both current production and disabled waste dams has been stored in a fully waterproof and sealed monitored landfill. Although many of these industrial residues have been removed and stored, there are still many traces of contamination in the soil and alluvial plain in the vicinity of the industrial area, and in the sediments accumulated in the banks and bed of the Consciência creek, a narrow and shallow tributary of the São Francisco River [2,7,8]. Previous studies carried out by this team [1,7,9–11] revealed high levels of sulphates (from the chemical oxidation of sulphide ores) and potentially toxic elements (PTEs) such as Zn, Pb, Cu, As, and Cd, from the surface to the deepest layers of the alluvium, sediments, and soils. In addition to the excessive concentrations, metals and As occur preferably in soluble and/or adsorbed forms in different mineral phases and most of them are transported through the sedimentary materials of Consciência creek and may even be mobilized through the bed of São Francisco River, reaching the opposite bank.

Given the high levels of these elements in labile forms, the removal of materials by dredging had to be discarded as an ex situ remediation technique [7,12]. This work aims to test the effectiveness of an in situ remediation methodology based on the immobilization of these PTEs through the application of zero-valent iron nanoparticles (nZVI). Zero-valent iron (Fe0) has been reported as a successful remediation agent for environmental issues and they are extensively used in soils, e.g., [13–16], due to its physical and chemical properties, namely high reduction power and non-toxicity. Its application reduces the potential leachability of contaminants, preventing their transport into deeper soil layers, rivers, and groundwater [17]. ZVI appears to be one of the most cost effective because it is cheaply available in large quantities as an industrial by-product, unlike other Fe-containing compounds [13,18,19].

At a nanoscale, these particles represent a new generation of effective environmental remediation technologies that may offer unique solutions for the removal or degradation of various chemical pollutants in contaminated groundwaters and soils, including chlorinated organic contaminants, transition metals, and metals in solution [14,16,20–27]. Zero-valent iron nanoparticles (nZVI) are more effective than macroscale ZVI (mZVI), iron powder, or iron filings under similar environmental conditions [14]. Due to small size in the range of several tens of nanometers, these ultrafine particles have a very high surface area and a higher number of reactive sites for the reduction and consequent adsorption of metals, which leads to extraordinary reduction capabilities, making them extremely reactive with a broad spectrum of toxic substances and potentially mobile, depending upon the hydrogeologic and geochemical conditions. They are highly applicable in groundwater remediation and wastewater treatment and can be used for in situ treatment by direct injection in soil or sedimentary deposits [28,29]. The injected particles migrate through soils and sediments until the depth where contamination was delimited. This physicochemical reactivity is also enhanced by its high porosity [29].

Many laboratory and pilot studies carried out, especially in the United States and Europe, have reported the high effectiveness of these nZVI-based technologies for the remediation of groundwater and contaminated soils [14,30–36]. However, results are often based on a limited contaminant target and there is a large gap concerning (1) the behavior and efficiency of nZVI in Fe-rich materials in tropical climates; (2) the behavior and efficiency of nZVI in aqueous metal-contaminated environments, for example, alluvial or stream sediments [37]; (3) the in situ immobilization of a wide range of target metals, most of them at very high concentrations; and (4) the in situ immobilization of metals with high ionization potential, such as Cd or Zn [15,38,39].

This study is an experimental test, at a laboratory scale, using nanoparticles of zero-valent iron (nZVI) in different types of soils and alluvial sediments from this highly con-

taminated industrial area. The aim of this study, besides being an attempt to solve a serious environmental problem, is to fill the gap that exists regarding the application of these particles in the decontamination of fine-grained alluvial sediments and soils, with high proportions of a wide range of metals mostly in soluble or readily soluble forms, in a tropical region dominated by Fe-rich lithologies.

In addition to the analysis of the geochemistry of the soluble phase of samples obtained from the laboratory scale tests, the effectiveness of this in situ remediation technique was assessed through a multivariate spatial approach applied to the variation of the contamination degree of the target PTEs in the samples, before the injection with nZVI and at the end of the laboratory-scale experiment. Indeed, the spatial quantification and hot-cluster definition for the selected pollutant metals allowed for the simulation of different distribution scenarios, using the metals' concentrations obtained in the sampled points [1]. Risk maps, broadly mentioned in literature, often use spatial pattern visualization of, e.g., pollutant concentration distribution, exposure and its effects, and vulnerability assessment; therefore, constituting a very powerful tool to support policymaking in a complex environmental risk assessment framework [40,41]. Moreover, stochastic approaches have been widely used to provide pivotal information for mitigation strategies [42]. The contents of As, Pb, Cu, Cd, and Zn, obtained after the application of nZVI in selected points, were projected overlapping each elements' risk map, thus allowing for the evaluation of the laboratory technique's accuracy in immobilizing the studied metals. A map is always a simplification of reality [40] offering a two-dimensional visualization and gathering and displaying the values of a limited number of variables [43].

## 2. Geographical and Geological Characterization

The metallurgical unit in study is located in Minas Gerais State (Brazil) (Figure 1a), next to the southern margin of the São Francisco River, and has an approximate area of 3 km$^2$. The nearest city is 3 km away (Três Marias) and is about 4 km from a large dam (Três Marias dam). It is bounded by two streams: to the north by Consciência Stream and its tributary (Grota Seca creek), and to the south by Barreiro Grande Stream, both tributaries of the São Francisco River.

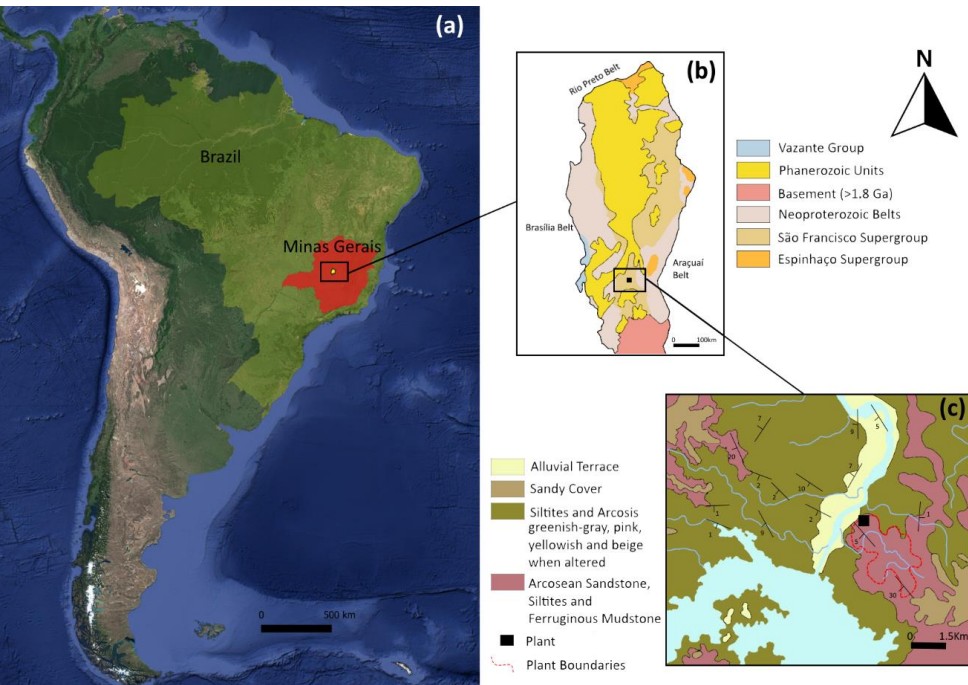

**Figure 1.** (**a**) Geographical location of the study area located in Minas Gerais state, South Brazil; (**b**) São Francisco Craton and its subdivisions; (**c**) geological setting and delimitation of the plant's area of influence [2,3].

From a geotectonic point of view, this area is inserted in the southern zone of the São Francisco Craton (Figure 1b), defined as one of the portions of the South American platform individualized by the orogenic processes of the Brasiliano Event (±600 Ma) [4,44,45]. This Craton is surrounded by Neoproterozoic Belts, the Brasília, and Rio Preto Belts to the west, and the Araçuaí Belt to the east. Near the city of Três Marias, the structural lines constrain the local morphology, exercising a strong control in the arrangement of the drainage network [3,5,6].

From a geological perspective, the study area is part of the Três Marias formation, which belongs to the Bambuí Group. Três Marias Formation, with an approximate age of 600–650 Ma, is a thick sedimentary sequence with relative lateral persistence, represented by layers of arkosic sandstones with fine to very fine grain size, intercalated by greenish-gray and violet arkosic siltites, rich in quartz, feldspar, and heavy minerals, such as Fe oxides, tourmaline, zircon, epidote, and garnet [2,5,46–48]—Figure 1c.

The depositional aspects of the Três Marias Formation are mainly defined by an erosive discordant nature, with no significant deformations, which attests to the relative tectonic stability of the central compartment during deposition [3,48].

The sedimentation process of Três Marias Formation suggests a depositional environment typical of a shallow continental shelf, dominated by storms and exposed to cyclic depositional events in which all the cycles describe periods of alternate sedimentary acreage during calm and stormy periods [3,6,46]. The pelitic and sandy deposits of Três Marias Formation identified in the area, indicate that the paleoenvironment, marked by depositional alternations, favored the intercalation of fine to very fine sediments, promoting the high degree of compression acquired [3].

The characterization of the aquifers was essentially based on the litho-structural aspects and on the nature and permeability of the rocks. Once the area is represented by horizontal and sub-horizontal sandstones and tabular siltites, water circulation and infiltration occur along with fracture systems, percolating through weathered rocks and recharging through overlapping granular systems [49]. These mechanisms of water circulation and storage create a fissured aquifer system, which is rather vulnerable to contamination.

## 3. Materials and Methods

In this study, we tested the nZVI efficiency at a laboratory scale, through the injection of different concentrations of nZVI solution, in three selected samples from the plant's area of influence, consisting of one soil (CA1-33) and two sediments from an extensive floodplain (CA2-10, CA3-18), with high metal contents, and a few of them over the toxic limit. The accuracy of this new in situ remediation technique was assessed through a multivariate spatial approach, based on the variation of levels of the extractable fraction of the target metals in samples, attained before and at the final step after the injection with nZVI. The metal contents corresponding to the end of the laboratory-scale experiment were projected overlapping the previously computed risk maps [1].

### 3.1. Samples Selected for This Study

The selection of samples to decontaminated with nZVI solution was based on data obtained in previous studies on the geochemical characterization and assessment of the contamination degree of soils and alluvial plains under the influence of this industrial unit [1,7,9,11] (Figure 2a). Data were used to compute a hundred simulated scenarios, through Sequential Gaussian Simulation (SGA), and the final Mean Image map (MI) allowed for identifying spatial risk clusters for metal contamination [1]. The spatial distribution patterns of the PTEs were established using a two-step geostatistical modeling methodology: (1) structural analysis and experimental variograms were carried out on each selected attribute (As, Pb, Cu, Cd, and Zn), and (2) Sequential Gaussian Simulation (SGS) was employed as a stochastic simulation algorithm. SGS begins by setting the univariate value distribution by performing a normal score transformation of the original values into a standard normal distribution. Normal scores at grid node sites were sequentially simulated

with simple kriging (SK) using normal score data and a zero mean [50] After all normal scores were simulated, they were retro converted to original rank values. For the calculus, the Space-Stat V software 4.0.18, Biomedwere, was used [51].

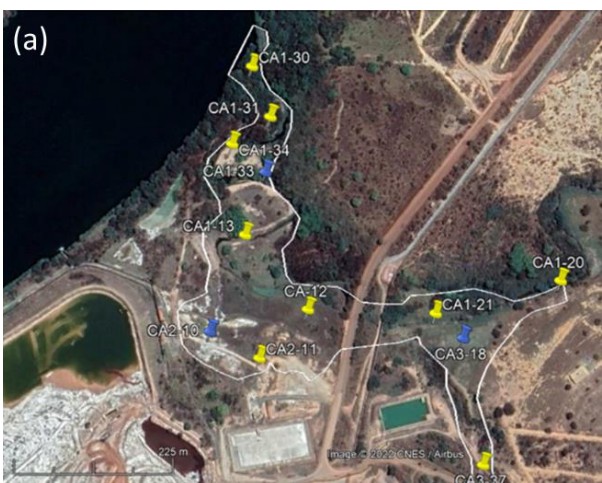 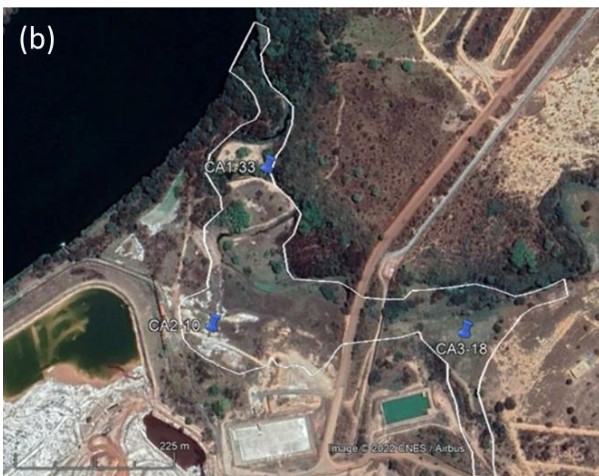

**Figure 2.** (**a**) Location of samples analyzed for geochemical characterization in the plant's area of influence; (**b**) test samples' location for zero-valent iron nanoparticles (nZVI) technology assessment.

The three test samples were selected (Figure 2b) according to the critical contents of metallic elements, the representation of the two main types of contaminated deposits in the area (soils and alluvial sediments), and the geographical distribution as they are influenced by different activities of the plant and tailing deposits. About 10 kg of the sample was collected, from the surface to a depth of 40 cm. The three selected samples correspond to (1) a red latosol, with critical contents of Zn and Pb—CA1-33; (2) an alluvial sediment from an extensive alluvial plain of Consciência creek, with critical contents of Zn, Pb, Cu, Cd, and As—CA2-10; and (3) alluvial sediment from the mouth of Grota Seca creek (Consciência's tributary) with high concentrations of Cd, As, Zn, and Pb—CA3-18.

The transport of the samples was made under refrigerated conditions (at about 4 °C) to preserve their original chemical conditions. Sampling was followed by immediate readings of pH, using a portable multi-parameter Consort, C5020, SP10T model probe (Thermo Fisher Scientific Inc., Brussels, Belgium). The evaluation of the contamination degree was performed in partially digested samples with hydrogen peroxide in open containers overnight and aqua regia ($HNO_3$ + HCl) in a high-pressure microwave digestion unit (Anton Paar Multiwave PRO, Graz, Austria) following the US EPA Method 3051A [52]. Metals and As contents were analyzed by optical emission spectroscopy with inductive plasma source (ICP-OES, Perkin-Elmer OPTIMA 8300, Mundelein, IL, USA), operated under the following conditions: plasma gas flow—10 L/min; auxiliary gas flow—0.2 L/min; PFA-ST3 Microflow atomizer 0.1–3 mL/min; nebulizer gas flow—0.70 L/min; sample flow—1.50 mL/min; RF Power—1450 watts; viewing modes—radial and axial; reading time—2–5 min; read delay—80 s; normal resolution; internal standard—yttrium. The accuracy and analytical precision of all the analyses were checked by the analysis of reference materials and duplicate samples in each analytical set.

### 3.2. Solution of Zero-Valent Iron Nanoparticles (nZVI)

In this study, it was used a highly reactive solution of zero-valent iron, NANOFER 25S® (NanoIron s.r.o., Rajhrad, Czech Republic), which is an aqueous dispersion of Fe(0) nanoparticles stabilized by a biodegradable organic and an inorganic modifier. These nanoparticles have a dimension lesser than 50 nm, a specific surface higher than 25 mg²/g, a specific density between 1.15 and 1.25 g/cm³ (20 °C), and a null surface charge (Fe(0)). Chemically, this material has a high content of iron in the range of 80–90 wt.% under the form of elemental iron (Fe) and magnetite ($Fe_3O_4$), also having carbon (C), in a water

solution with an organic stabilizer. According to the producer† data, the NANOFER 25S®
has the following composition, summarized in Table 1.

**Table 1.** Specifications of zero-valent iron nanoparticle solution used in the test (NANOFER 25S®,
NanoIron s.r.o., Rajhrad, Czech Republic).

| Chemical Composition of $Fe^0$ Nanoparticles | Fe (Core), FeO (Capsule) |
|---|---|
| Mass percentage of the solution | 20% |
| $Fe^0$ mass in solid fraction | 80% |
| Other substances of the solid fraction | $Fe_3O_4$, FeO, C |
| Other substances in the liquid fraction | Organic Stabilizer |
| Particle shape | Spherical |
| $Fe^0$ particle size | D50 nm < 50 |
| Specific surface | >25 $m^2/g$ |
| Color | Black |
| Solution density | 1210 $kg/m^3$ |
| $Fe^0$ Density | 7870 $kg/m^3$ |
| $Fe_3O_4$ Density | 5700 $kg/m^3$ |

*3.3. Procedure and Implementation of the Column Laboratory Test*

The test was carried out with four replicates of each sample, by injecting throughout
a peristaltic pump, deionized water (blank), and the same volume of nZVI solutions
with three different concentrations, prepared from the standard solution (NANOFER
25S–240 g/L), just before the injection, so that reactivity is not lost:

- Blank—Deionized water
- Nanoparticles with 1 g/L concentration
- Nanoparticles with 3 g/L concentration
- Nanoparticles with 7 g/L concentration

The samples were placed on transparent acrylic columns with an internal diameter
of 4.6 cm, fixed on a support. Each column was air-sealed through silicone stoppers.
The bottom stopper was perforated to introduce a PVC tube connected to a tap that
plugged into the peristaltic pump tubes. The other extremity of the peristaltic pump tube
was connected to a silicone tube, which was inserted into the container with the nZVI
solution to be injected. This tube was wide enough to allow for the flow of the nanoparticle
solution, but it could not be too broad to avoid the sedimentation of the high-density
particles. Each column had two holes, at the top and at the bottom, that were plugged with
small silicon stoppers designed for the introduction of rhizome samplers for the extraction
of interstitial water. At the bottom and top of the soil/sediment column, it was placed
washed gravel to ensure good drainage during the test (Figure 3).

Samples were placed in a saturated condition in beakers with distilled water for one
week, to reach a balance between the sedimentary particles, the pre-existing interstitial
water, and the water ascended by capillarity. For each sample, the amount of water needed
for saturation was quantified, and based on this volume and on the percentage of initial
moisture, the total amount of water retained was calculated. This value was used to
determine the volume of nanoparticles solution to be injected. An injection volume of
250 mL of each solution and deionized water (for the blank) was estimated for each sample,
replicated over four columns.

After the saturation period, and in a hermetically sealed environment to avoid any
oxygen entry, the columns were placed on supports (Figure 3a), and the injection of the
solutions (in continuous agitation to avoid sedimentation), with the aid of a peristaltic
pump, was made through the bottom to nullify the effect of gravity (Figure 3c). As they
were injected, the nZVI solutions slowly ascended in the columns, opening cracks in the
sedimentary material, and occupying the pre-existent water spaces in the pores, expelling
it to the top of the tubes (Figure 3d). Part of this water was collected for analysis.

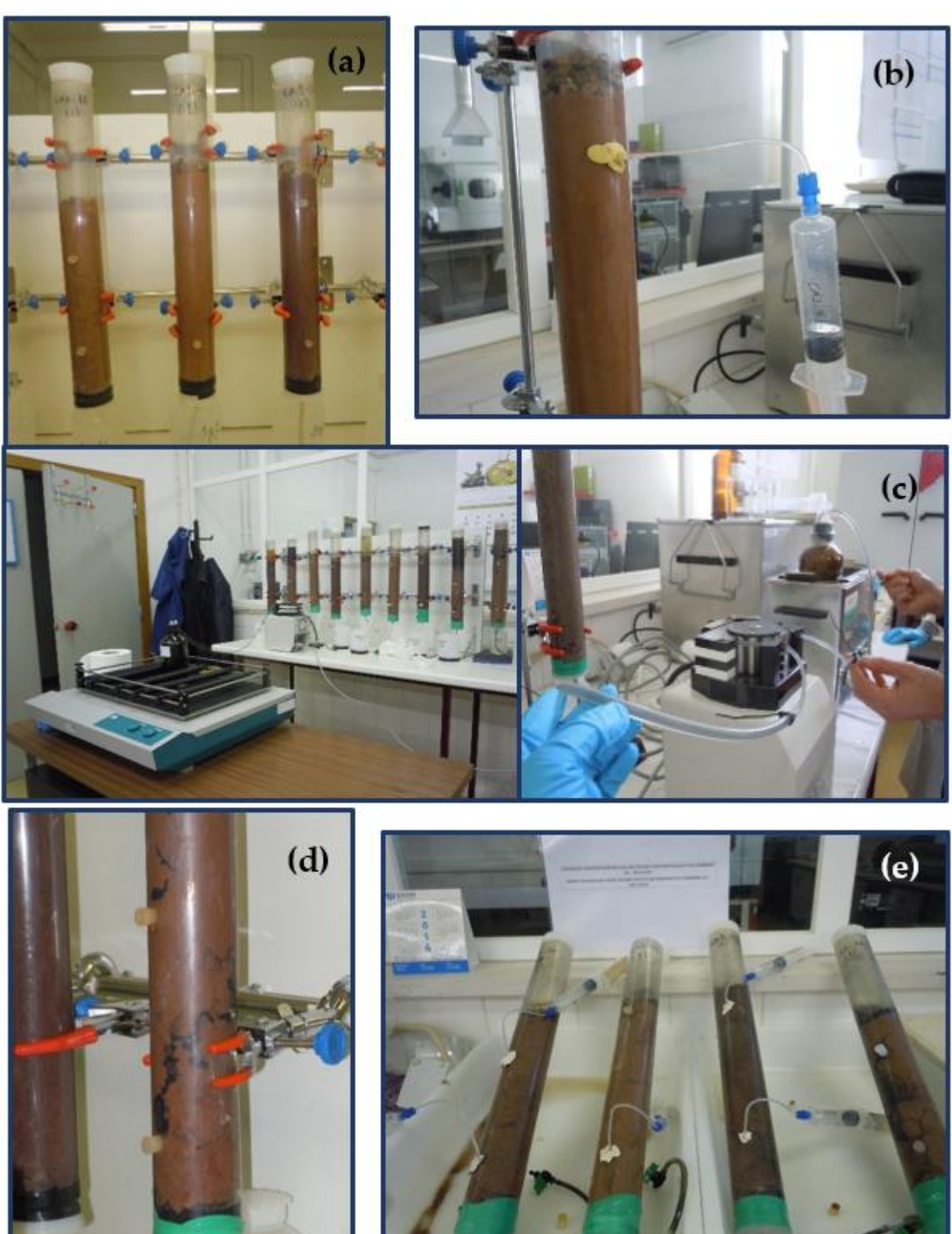

**Figure 3.** Steps of the injection test with zero-valent iron nanoparticle (nZVI) in metal-contaminated sedimentary materials: (**a**) columns with the saturated samples, (**b**) extraction of the interstitial water before injection, (**c**) injection of the nZVI solutions (at three different concentrations) and deionized water (blank) into the sample columns, (**d**) appearance of the samples after injection of the nZVI solution, and (**e**) extraction of the interstitial water after nZVI injection and for various time periods, from 24 h after injection, up to 4 months.

To check the effect of the nanoparticles in the decreasing of the metal contents in the soluble phase of the soil and sediments, interstitial water samples were extracted (using rhizome samplers) at the top and bottom of each column, before (Figure 3b) and after injection at regular time periods (Figure 3e). In each water sample, it was measured pH, Eh, and conductivity (benchtop multi-parameter Consort, C863 model, probes (SP10T model for pH, SP50X model for redox potential, SK24T model for conductivity), followed by acidification with $HNO_3$ for subsequent geochemical analysis of metals by optical emission spectroscopy with inductive plasma source, using the same equipment and under the same operating conditions as for the analysis of soil and sediment samples.

The accuracy and analytical precision of all the analysis were checked by the analysis of duplicate samples in each analytical set.

The periods of interstitial water extraction were as follows:

1st—Before the injection of the solution of nZVI;
2nd—After 24 h;
3rd—After 48 h;
4th—After 72 h;
5th—After 1 week;
6th—After 2 weeks;
7th—After 1 month;
8th—After 2 months;
9th—After 4 months.

After 4 months of testing, the soil and alluvial samples were removed from the columns, homogenized, and in each composite sample, an interstitial water sample was collected. Solid samples were then digested following the same procedure described in Section 3.1. The main objective was comparing the contamination degree of the target metals in the samples before and after the injection with nZVI.

## 4. Results

*4.1. Geochemical Characterization of the Samples Used for Testing nZVI at a Laboratory Scale, and Evaluation of the Contamination Degree of the Industrial Area*

The geochemical analyses of the selected samples for the lab test are presented in Figure 4. In Table 2, the reference values used to evaluate the quality of sediments and alluvium are presented. These values followed the CONAMA Resolution, No. 454, 2012, which establishes the guiding values for the evaluation of the quality of dredged sediments.

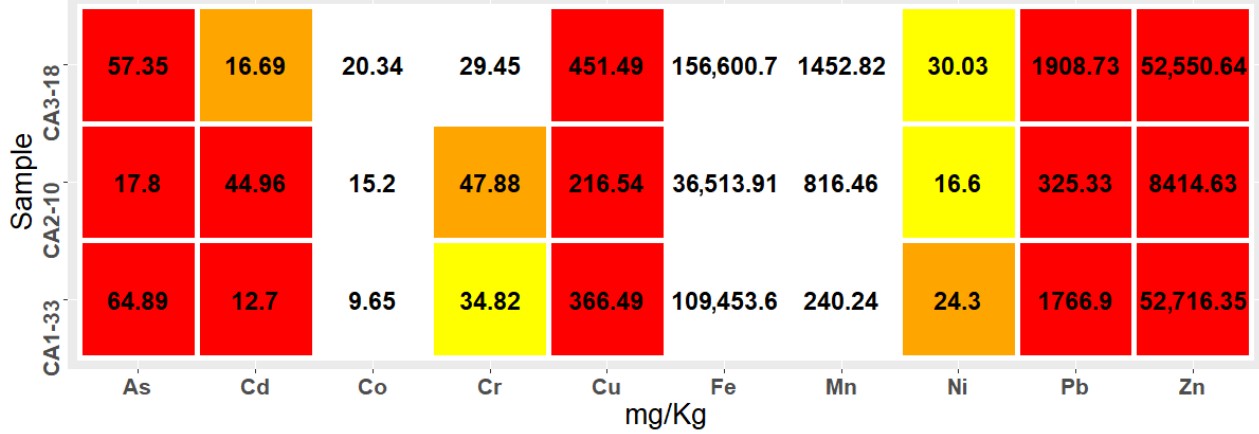

**Figure 4.** Heat map of the concentrations of the extractable forms of the metallic elements (by *aqua regia* digestion; evaluation of the degree of contamination) of the samples used in the laboratory assay. Color label: red—above the critical content; orange—caution content; yellow—content above the quality limits for soils [53] and sediments [54].

**Table 2.** Reference values for the quality of sediments and alluvium, according to guideline values established by legislation for the quality of dredged sediments (CONAMA Resolution, No. 454, 2012).

| Sediments and Alluvium (CONAMA, 2012) | As (mg kg$^{-1}$) | Cd (mg kg$^{-1}$) | Cr (mg kg$^{-1}$) | Cu (mg kg$^{-1}$) | Ni (mg kg$^{-1}$) | Pb (mg kg$^{-1}$) | Zn (mg kg$^{-1}$) |
|---|---|---|---|---|---|---|---|
| Normal | <5.9 | <0.6 | <33.0 | <17.0 | <14.0 | <8.4 | <58.0 |
| Intermediate | | | 33.0–37.3 | 17.0–35.7 | 14.0–18.0 | 8.4–35.0 | 58.0–123.0 |
| Caution | 5.9–17.0 | 0.6–3.5 | 37.3–90.0 | 35.7–197.0 | 18.0–35.9 | 35.0–91.3 | 123.0–315.0 |
| Critical | >17.0 | >3.5 | >90.0 | >197.0 | >35.9 | >91.3 | >315.0 |

Considering the elements which represent the majority in the ores processed in the plant, Zn, Cd, Cu, As, and Pb, the levels found in these samples, representative of soils and alluvial sediments of the affected area, are extremely high, most of them above the limits considered toxic, according to the Brazilian legislation for soils [53] and sediments [54]. The pH values range between near acidic (5.77 in CA2-10) to near neutral (6.33 for CA3-18 and 6.54 for CA1-33).

According to previous studies [1,7,9], in addition to the excessive concentrations from the surface to the deepest layers (80–100 cm), they occur in easily mobilized phases (soluble, exchangeable cations, associated with Mn-oxides) and spread rapidly to areas further from the source of contamination, thus representing the elements with the greatest environmental hazard. In these three samples, 80–100% of Cd and Zn occur in these more labile forms, whereas Pb and Cu have a lower solubility and more heterogeneous distribution (Pb: 25% in CA2-10, 60–70% in CA3-18, and 45–55% in CA1-33; Cu: 10–20% in CA2-10; 60% in CA3-18, and 20–40% in CA1-33). Although As values exceed the critical values, this element has a heterogeneous behavior over the area of concern but, in general, only 20% occur in more mobile fractions.

*4.2. Data Obtained in the Laboratory-Scale Test of Injection of Nanoparticles of Zero-Valent Iron (nZVI)*

4.2.1. General Remarks

In this experiment a drought situation was simulated, since the system was closed, and the extracted water was not replaced. The interstitial water became more concentrated over time, but the material always remained moist. This corresponds to an extreme situation in which it was predicted that the decrease of the soluble forms of metals would be less than would occur in a real situation.

One of the key problems observed in this test was the poor diffusion of the nZVI suspensions during the injection process. The suspension, injected from the bottom of a column filled with a water-saturated sample, was able to perforate all the sample, reaching the top. However, the diffusion was not uniform and nuclei with aggregated particles were observed. Thus, the transport of the iron nanoparticles was slowed down due to rapid aggregation of them. This aggregation of non-surface-modified nZVIs has been an issue and a problem since the very first field demonstration performed in 2001 by Elliot and Zhang [55]. The rate of aggregation increased with growing concentration of nZVI particles and with growing ionic strength of the solution, although an attempt was made to homogenize the particles by stirring as well as possible before injection.

The data obtained over the 4 months of the laboratory batch studies correspond to the analysis of the interstitial water extracted from the upper and lower layers of the samples, in each selected period. The analysis includes physical-chemical parameters (pH, redox potential, conductivity) and geochemistry of the metals with levels of concern: Zn, Cd, Cu, Pb, and As.

4.2.2. Variation of the Physic-Chemical Parameters (pH, Redox Potential, Conductivity)

Table 2 shows the pH and Eh values of the interstitial water extracted in all the samples, in each selected period during the laboratory experiment with injection of nZVI.

- pH:

One of the main parameters affecting metal removal by zero-valent iron nanoparticles is pH [20,39,56–60]. The impact of pH on the metal's removal by nZVI depends on the oxidation state of the metal, the removal mechanism, and the characteristics of the outer layer of ZVIs.

According to most of the literature [18–20,38,39,56–61], nZVI are an important reducing agent that, in aqueous media, react with dissolved oxygen, water, and other oxidants (nitrates, sulphates). They transfer electrons to metals in solution, reducing them. The $Fe^0$, on the other hand, when transferring electrons, oxidizes, changing into iron oxides and hydroxides, releases hydroxyl radicals, as shown in Equation (1) [56,61], increasing the pH and decreasing the redox potential of the system:

$$2Fe^0 + O_2 + 2H_2O \rightarrow 2Fe^{2+} + 4OH^- \tag{1}$$

However, in this laboratory batch trial, the variation mentioned by the authors was only observed in the first 2 days of the nZVI injection. From the third day of the experiment the opposite occurred, the redox potential increases following the decrease of pH, although keeping similar relation between both (Table 3). This shift in pH and redox potential variation has been observed by several authors, e.g., [55], who state that over time there is a gradual decline of pH and increase of Eh, trending back to the starting conditions.

**Table 3.** pH and Eh values of the interstitial water extracted from the upper and lower layers of the samples, in each selected period, for the different concentrations of nZVI solution.

| ZVI | Layers | pH | | | | | | | | | Eh (mV) | | | | | | | | |
|---|---|---|---|---|---|---|---|---|---|---|---|---|---|---|---|---|---|---|---|
| | | $t_0$ | $t_{24H}$ | $t_{48H}$ | $t_{72H}$ | $t_{7d}$ | $t_{14d}$ | $t_{30d}$ | $t_{60d}$ | $t_{120d}$ | $t_0$ | $t_{24H}$ | $t_{48H}$ | $t_{72H}$ | $t_{7d}$ | $t_{14d}$ | $t_{30d}$ | $t_{60d}$ | $t_{120d}$ |
| | | | | | | | **CA1-33** | | | | | | | | | | | | |
| 1 g/L | Upper | 4.44 | 6.61 | 6.90 | 6.53 | 2.70 | 6.22 | 3.20 | 3.91 | 5.87 | 554.0 | 476.6 | 276.2 | 287.9 | 464.4 | 253.3 | 384.3 | 685.4 | 305.4 |
| | Lower | 4.07 | 6.76 | 6.32 | 6.26 | 6.09 | 5.62 | 4.76 | 3.27 | 5.65 | 431.2 | 424.9 | 272.8 | 466.0 | 296.4 | 233.8 | 375.4 | 639.7 | 321.6 |
| 3 g/L | Upper | 3.50 | 6.88 | 7.10 | 6.72 | 4.11 | 3.00 | 3.33 | 3.23 | 4.77 | 417.4 | 169.4 | 187.8 | 184.6 | 346.3 | 444.7 | 398.7 | 449.1 | 264.8 |
| | Lower | 6.52 | 6.97 | 7.04 | 6.57 | 3.64 | 3.37 | 3.22 | 6.38 | 6.39 | 354.4 | 163.2 | 193.3 | 160.5 | 383.5 | 411.0 | 412.3 | 131.1 | 161.6 |
| 7 g/L | Upper | 5.79 | 6.89 | 6.79 | 2.54 | 2.56 | 3.06 | 5.79 | 6.34 | 6.78 | 235.4 | 141.4 | 160.9 | 486.6 | 458.8 | 422.6 | 223.6 | 124.4 | 317.7 |
| | Lower | 6.71 | 6.17 | 7.05 | 2.13 | 2.07 | 3.25 | 4.43 | 6.42 | 6.60 | 233.9 | 102.5 | 160.0 | 506.0 | 444.6 | 462.8 | 288.4 | 121.1 | 305.2 |
| 0 g/L | Upper | 6.70 | 6.48 | 2.94 | 3.05 | 2.50 | 3.41 | 2.85 | 3.61 | 4.61 | 245.7 | 204.8 | 277.7 | 496.7 | 683.2 | 485.5 | 567.8 | 157.6 | 301.5 |
| | Lower | 6.65 | 6.73 | 3.01 | 4.72 | 5.96 | 4.18 | 3.51 | 3.19 | 4.56 | 231.5 | 207.8 | 290.5 | 449.4 | 610.6 | 477.0 | 539.4 | 166.3 | 301.5 |
| | | | | | | | **CA2-10** | | | | | | | | | | | | |
| 1 g/L | Upper | 5.21 | 5.84 | 5.07 | 2.76 | 3.91 | 3.00 | 2.95 | 3.35 | 3.58 | 284.7 | 99.7 | 206.7 | 382.9 | 322.3 | 369.2 | 402.2 | 359.3 | 189.0 |
| | Lower | 5.16 | 5.64 | 4.61 | 2.60 | 3.90 | 3.25 | 3.99 | 3.44 | 3.71 | 310.6 | 201.9 | 216.6 | 410.8 | 357.9 | 409.8 | 418.7 | 450.8 | 234.8 |
| 3 g/L | Upper | 5.51 | 6.28 | 6.13 | 2.57 | 4.19 | 4.08 | 3.26 | 4.04 | 4.87 | 245.8 | 51.3 | 129.7 | 384.6 | 290.4 | 407.4 | 430.4 | 152.3 | 275.4 |
| | Lower | 5.23 | 5.93 | 4.91 | 2.45 | 1.84 | 3.98 | 3.42 | 2.95 | 4.07 | 234.8 | 76.4 | 173.8 | 390.1 | 439.7 | 436.0 | 458.2 | 286.0 | 380.2 |
| 7 g/L | Upper | 5.94 | 6.10 | 6.23 | 2.63 | 3.08 | 3.02 | 2.98 | 3.21 | 4.76 | 261.6 | 129.3 | 133.7 | 395.3 | 314.2 | 383.4 | 471.7 | 281.5 | 397.3 |
| | Lower | 5.91 | 5.79 | 6.23 | 2.25 | 2.53 | 3.77 | 3.25 | 4.38 | 5.10 | 229.7 | 147.0 | 167.2 | 457.4 | 399.1 | 417.4 | 489.2 | 128.1 | 321.5 |
| 0 g/L | Upper | 2.57 | 2.93 | 2.72 | 2.77 | 2.72 | 3.20 | 3.05 | 3.63 | 3.54 | 226.2 | 551.8 | 545.3 | 643.1 | 627.0 | 593.9 | 181.6 | 569.1 | 320.7 |
| | Lower | 2.57 | 3.64 | 4.78 | 3.45 | 3.06 | 3.07 | 3.70 | 3.60 | 3.26 | 331.0 | 486.5 | 503.4 | 579.6 | 584.1 | 524.0 | 202.0 | 574.1 | 336.5 |
| | | | | | | | **CA3-18** | | | | | | | | | | | | |
| 1 g/L | Upper | 4.31 | 6.56 | 6.78 | 2.55 | 4.54 | 2.74 | 3.73 | 6.29 | 5.21 | 360.4 | 170.7 | 206.0 | 423.7 | 296.3 | 392.9 | 418.9 | 306.0 | 388.8 |
| | Lower | 5.79 | 6.73 | 6.87 | 2.87 | 5.51 | 3.19 | 3.08 | 3.61 | 6.65 | 338.8 | 179.4 | 213.2 | 475.2 | 281.6 | 478.5 | 488.9 | 442.4 | 374.1 |
| 3 g/L | Upper | 5.81 | 6.65 | 6.91 | 2.19 | 4.33 | 4.30 | 2.93 | 6.38 | 5.92 | 348.3 | 133.2 | 159.1 | 486.9 | 387.3 | 336.0 | 417.3 | 121.0 | 305.3 |
| | Lower | 6.71 | 6.86 | 7.07 | 2.40 | 3.37 | 3.74 | 3.71 | 6.55 | 6.69 | 331.3 | 138.1 | 159.5 | 489.6 | 455.0 | 419.6 | 400.8 | 125.9 | 322.8 |
| 7 g/L | Upper | 5.27 | 6.65 | 6.97 | 2.24 | 2.47 | 3.08 | 3.79 | 5.19 | 6.90 | 272.6 | 63.9 | 146.7 | 453.8 | 445.1 | 380.9 | 395.8 | 130.5 | 271.4 |
| | Lower | 6.04 | 6.79 | 6.95 | 2.28 | 2.68 | 3.52 | 3.13 | 6.37 | 6.98 | 255.4 | 95.1 | 148.6 | 456.7 | 458.9 | 429.7 | 502.2 | 127.4 | 262.3 |
| 0 g/L | Upper | 4.65 | 6.69 | 5.22 | 2.36 | 3.75 | 3.48 | 3.41 | 3.90 | 3.87 | 343.0 | 212.6 | 258.3 | 527.0 | 537.8 | 521.7 | 528.4 | 574.9 | 543.7 |
| | Lower | 6.43 | 6.70 | 2.32 | 2.69 | 4.19 | 3.15 | 3.31 | 3.77 | 3.52 | 298.0 | 196.3 | 448.6 | 521.6 | 459.9 | 540.6 | 531.0 | 579.6 | 568.2 |

Moreover, there was a significant difference between the pH values of the samples measured in situ on the sampling day ($t_{is}$) and the pH values of the interstitial water measured at time zero of the laboratory experiment ($t_0$), which correspond to samples saturated for 7 days with distilled water. The latter values were significantly lower and more heterogeneous between replicates and, in each replicate, between the subsamples taken from the top and the bottom of the sedimentary column (e.g., CA1-33 (1): ($t_{ls}$) pH 6.54; ($t_0$) upper: pH 4.44, lower: pH 4.07; CA1-33 (2): ($t_{is}$) pH 6.54; ($t_0$) upper: pH 3.50, lower: pH 6.52).

For each concentration of the nZVI solution used in the experiments and for the blank, there was a change pattern of the pH values over time that was similar in all samples, but different between distinct concentrations. The temporal variation patterns of the blank and the solution with lower nZVI concentration were very similar, which may indicate that the main chemical reactions responsible for the pH changes should correspond to hydrolysis, overlapping the influence of nZVI action. For the higher concentrations, the pH increase observed at the beginning of the experiment may be due to the side reaction of iron from the nanoparticles that reduces water to hydrogen and hydroxide [21,22,55].

The uniformity of the pH variation over time was directly proportional to the suspension's concentrations. In the columns corresponding to the injection of the solutions with higher concentrations (3 and 7 g/L), there was an abrupt drop of the pH values after 2–3 days and a more gradual rise after 7–14 days, with a plateau after 60 days. The exception is the alluvium sample CA2-10, which kept low pH values until the end of the tests—120 days (see values in Table 3).

- Redox Potential:

In general, values for the different concentrations of the injected suspension were similar and followed approximately the same variations, especially in the samples corresponding to more concentrated suspensions (3 and 7 g/L). For all the samples and for all concentrations of nZVI, the oxidation conditions were higher up to the first 30 days of the test. This increase was followed by a sharp decrease until 60 days and a new increase until the end of the test, with values that tended to approach the initial ones.

- Conductivity

Conductivity values were only read up to day 60 due to lack of interstitial water required for analysis, at the end of the experiment. These values followed an oscillation throughout the trial period, very similar for all the concentrations of the nZVI suspension. In each sample and for each concentration, values were distinct between the top and the bottom of the column, given the insufficient dispersion of the nanoparticles throughout its length and the aggregation observed in various sectors, creating different environmental conditions. A sharp increase of conductivity was observed after the first day of nZVI injection, particularly for the most concentrated suspension (7 g/L) until the seventh day, which was the period from which a gradual decrease was observed (Figure 5). These variations were followed by variations of the metal's concentrations, which were also analyzed in the interstitial water.

### 4.2.3. Variation of the Immobilization Rate of the Target Contaminants

The main contaminants in the study area are zinc, lead, and cadmium, followed by copper and arsenic, so this work will only focus on these target metals. In the graphical representations of the variation of metals over the test period, to override the scale effect due to the high diversity of elements and their concentrations, the percentage of metals is represented in relation to their initial concentration before nanoparticle injection, which represents 100%.

For all the elements and similarly to what was observed for pH and redox potential, in each sediment column there was some discrepancy in the variation of metal content in the pore water extracted at the top and bottom, this variation being greater in the sample corresponding to the injection of the most concentrated suspension (7 g/L).

pH Effect

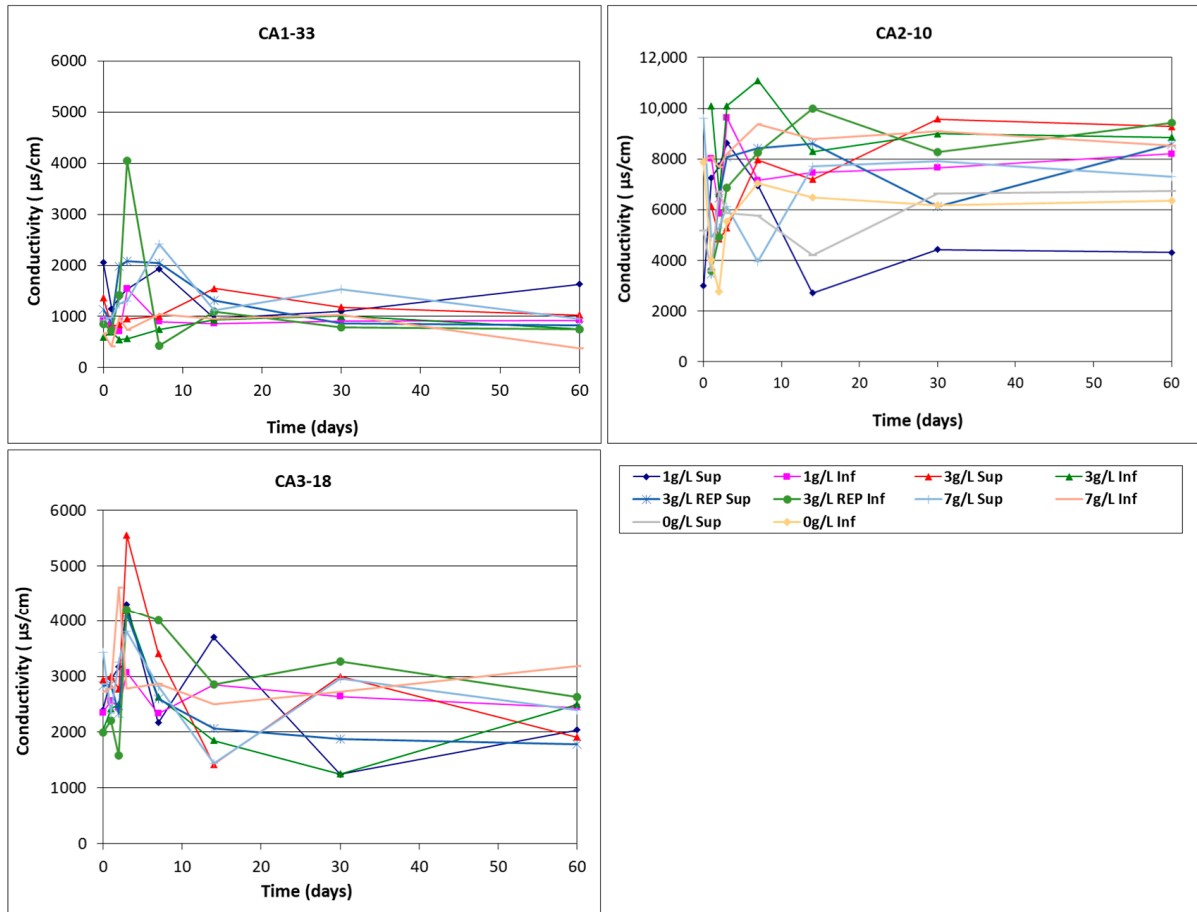

**Figure 5.** Variation of conductivity over the test period, at different concentrations of the nZVI suspension (blank-0 g/L, 1 g/L, 3 g/L, and 7 g/L).

In all the samples, the fluctuations of the retention rate of Zn and Cd by nZVI, followed the oscillations of pH values, increasing the levels of the soluble phase in more acidic conditions and decreasing them by increasing alkalinity.

After the injection of the nZVI, metals contents decreased in general, gradually during the first 2 days, increasing between the third and seventh day, which corresponds to the period with the lowest pH values (Figure 6, Table 3). After 7 days, the levels decreased again up to about 60 days, following the gradual increase of pH. These variations occurred both for elements with the mobility strongly dependent on pH, such as Zn and Cd, and for elements with other retention mechanisms, less pH dependent, such as Pb and Cu (Figure 7). Possibly due to the lower pH dependence of Pb mobility, its variation in the aqueous phase was more regular. This higher regularity was also observed throughout the columns, with Pb concentrations showing very similar patterns at the top and bottom of each column.

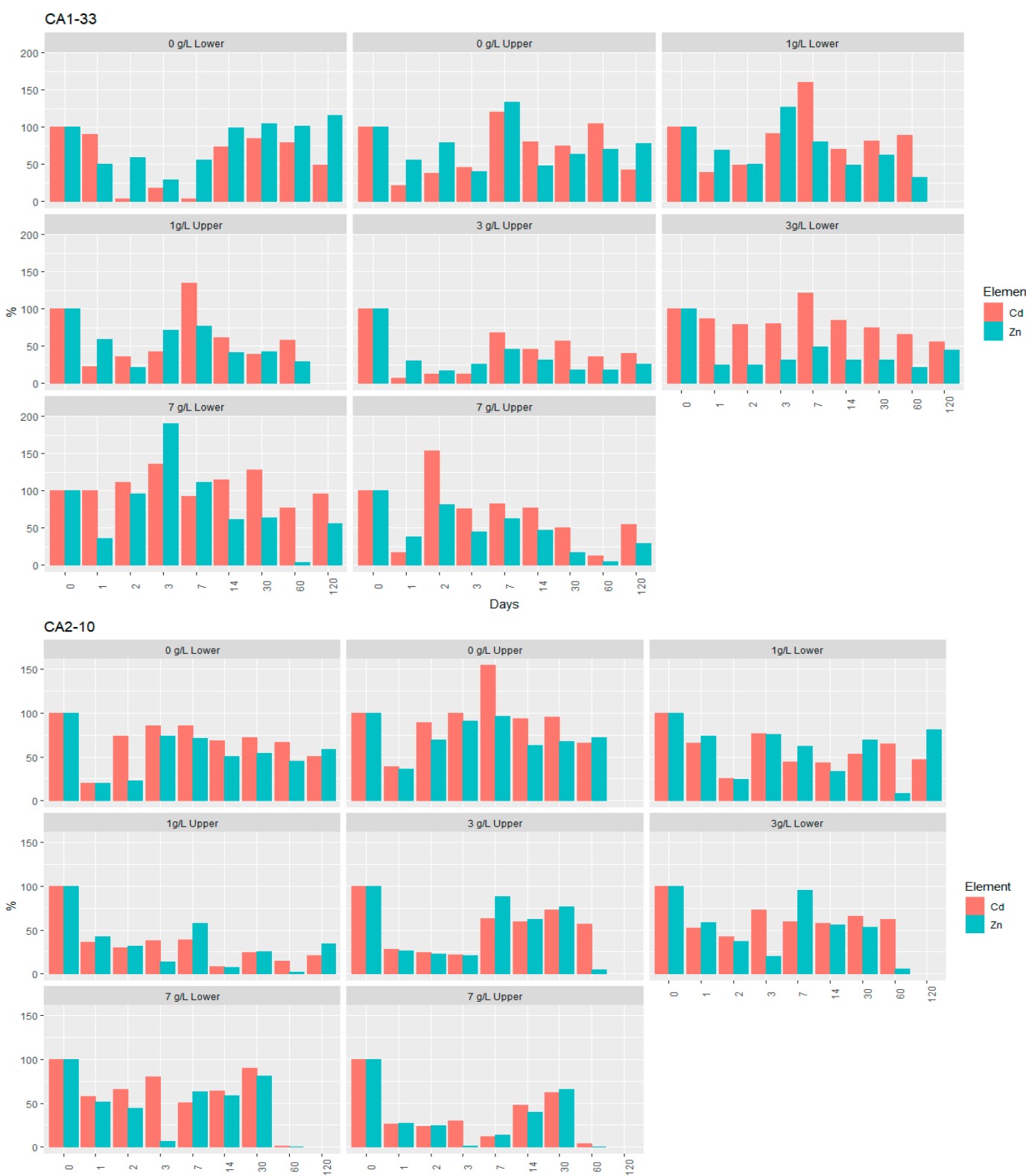

**Figure 6.** *Cont.*

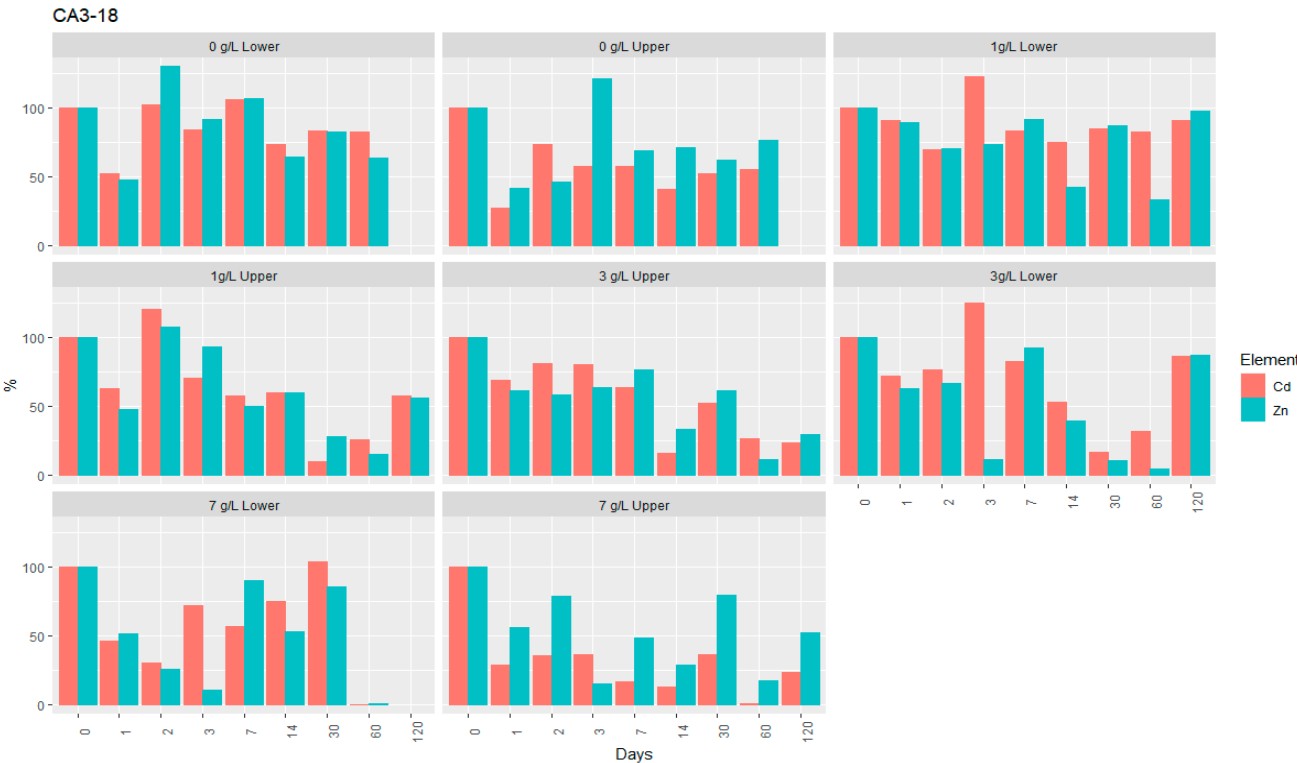

**Figure 6.** Changes in the Zn and Cd contents (%) in the soluble phase of samples, over 120 days of the remediation batch tests after the injection of nZVI suspension (concentrations of 0, 1, 3, and 7 g/L).

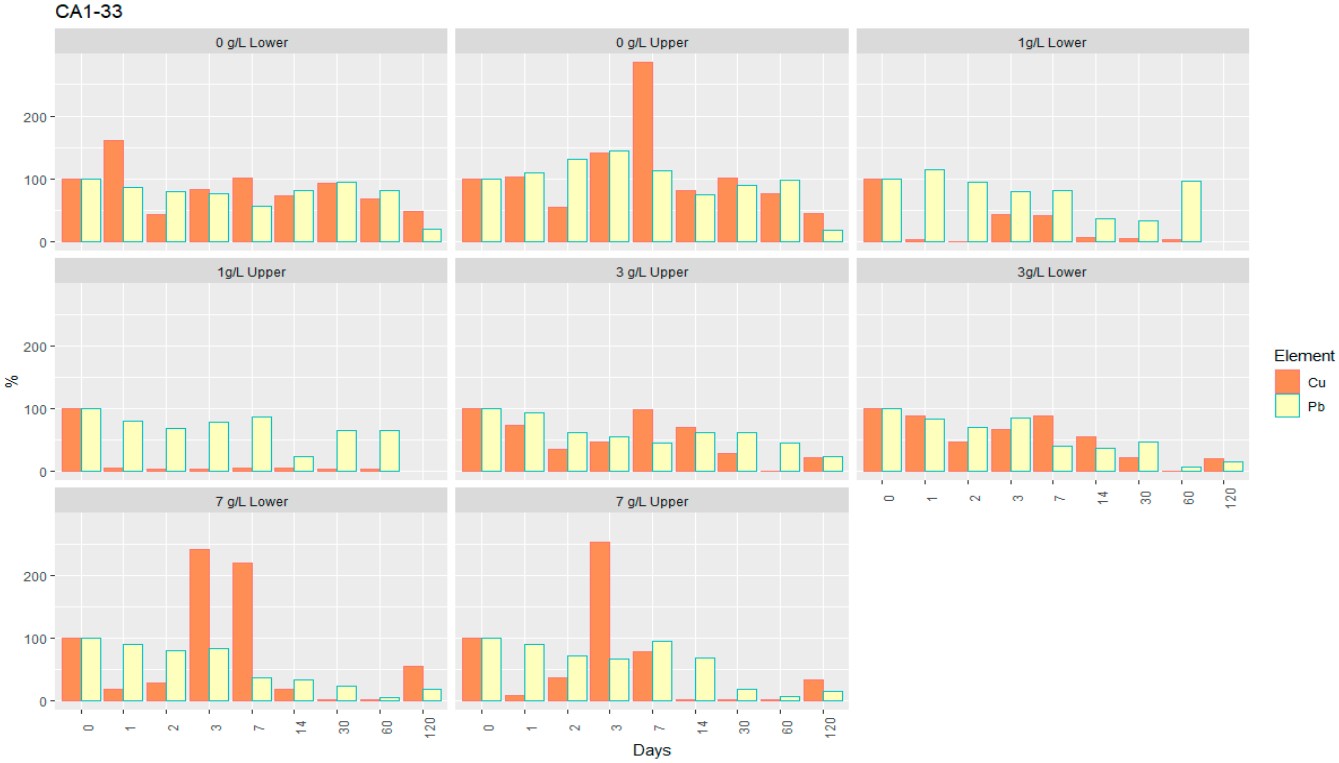

**Figure 7.** *Cont.*

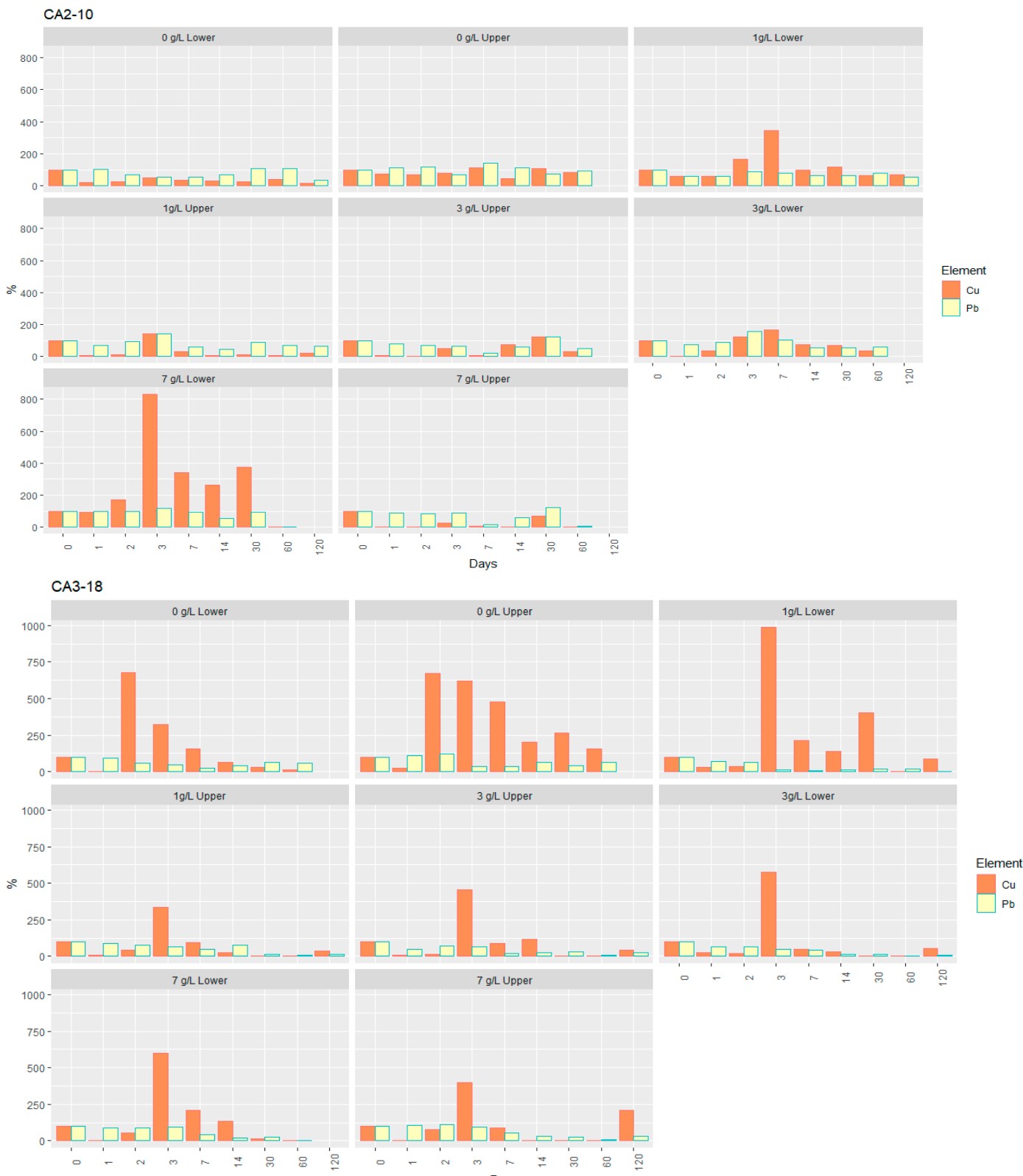

**Figure 7.** Changes in the Pb and Cu contents (%) in the soluble phase of samples, over 120 days of the remediation batch tests after the injection of nZVI suspension (concentrations of 0, 1, 3, and 7 g/L).

Effect of Suspension Concentration and Injection Time of nZVI

The effect of nZVI injection on the reduction of sediment and soil contamination levels, was assessed by two ways: (1) metal immobilization rate in the soluble fraction of the

samples, collected in two layers of the columns (upper and lower) throughout the various phases of the experimental test (Figures 6–8); and (2) comparison of metal contents in composite samples of interstitial water collected in homogenized samples (i) in the initial phase, before injection and placement of the samples in the columns, and (ii) in the final phase, after their removal (Table 4).

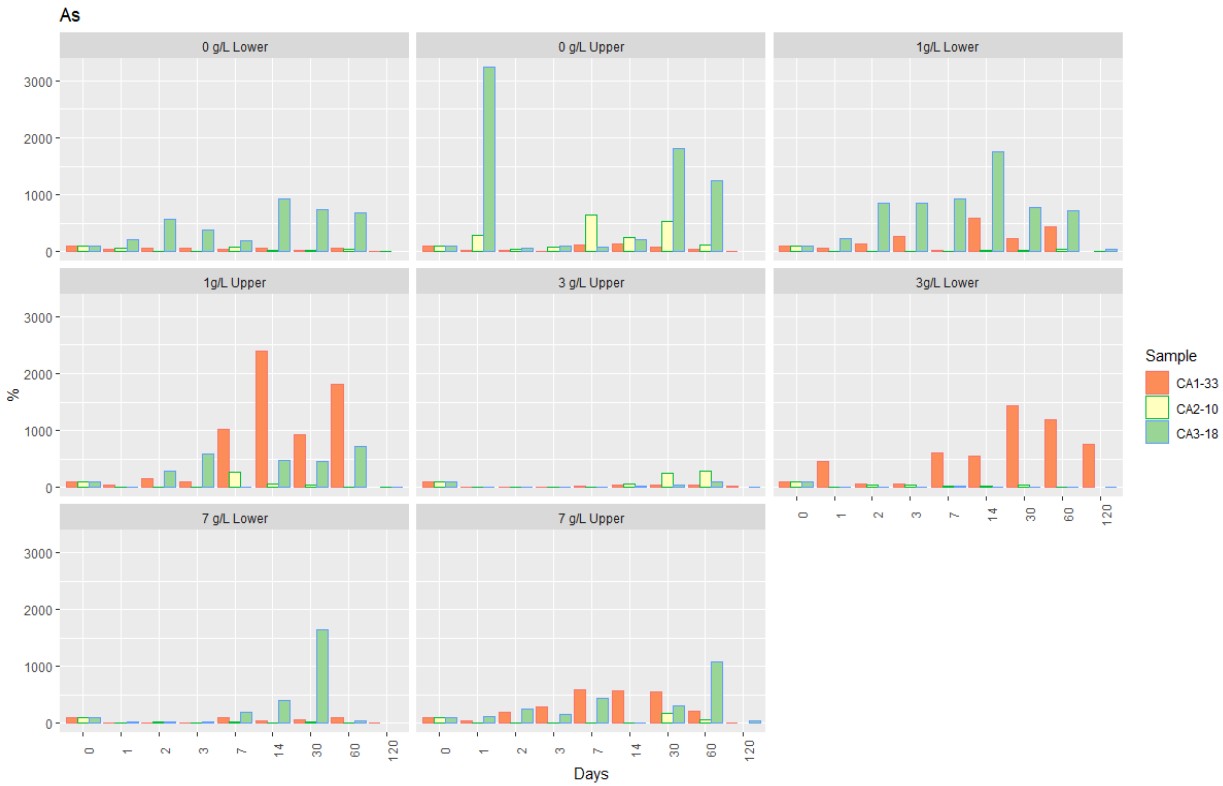

**Figure 8.** Changes in the As contents (%) in the soluble phase of samples, over 120 days of the remediation batch tests after the injection of nZVI suspension (concentrations of 0, 1, 3, and 7 g/L).

**Table 4.** Comparison between the initial contents of metals in the interstitial water of homogenized samples before injection of nZVI (t = 0) and their final contents, after 120 days (t = 120). For each element, values are given in concentrations (mg/L) and in % calculated as $[mg/L]_{t=120\ days}/[mg/L]_{t=0}$.

| ZVI | As (mg/L) | | As (%) | Cd (mg/L) | | Cd (%) | Cu (mg/L) | | Cu (%) | Pb (mg/L) | | Pb (%) | Zn (mg/L) | | Zn (%) |
|---|---|---|---|---|---|---|---|---|---|---|---|---|---|---|---|
| | t = 0 | t = 120 | t = 120 | t = 0 | t = 120 | t = 120 | t = 0 | t = 120 | t = 120 | t = 0 | t = 120 | t = 120 | t = 0 | t = 120 | t = 120 |
| | | | | | | **CA1-33** | | | | | | | | | |
| 1 g/L | 0.005 | 0.001 | 20 | 0.137 | 0.081 | 59 | 0.375 | 0.044 | 12 | 0.271 | 0.111 | 41 | 29.300 | 31.957 | 109 |
| 3 g/L | 0.055 | 0.001 | 2 | 0.110 | 0.041 | 38 | 0.102 | 0.028 | 27 | 0.304 | 0.031 | 10 | 24.001 | 8.852 | 37 |
| 7 g/L | 0.023 | 0.001 | 4 | 0.093 | 0.041 | 44 | 0.061 | 0.023 | 38 | 0.290 | 0.020 | 7 | 27.596 | 9.205 | 33 |
| 0 g/L | 0.061 | 0.001 | 2 | 0.115 | 0.040 | 35 | 0.064 | 0.019 | 29 | 0.235 | 0.041 | 18 | 22.023 | 17.001 | 77 |
| | | | | | | **CA2-10** | | | | | | | | | |
| 1 g/L | 0.057 | 0.003 | 5 | 4.568 | 0.813 | 18 | 0.420 | 0.123 | 29 | 1.388 | 0.750 | 54 | 950.101 | 309.307 | 33 |
| 3 g/L | 0.057 | 0.001 | 2 | 3.622 | 2.699 | 75 | 0.373 | 0.474 | 127 | 1.908 | 1.267 | 66 | 774.770 | 610.141 | 79 |
| 7 g/L | 0.062 | 0.001 | 2 | 3.808 | 2.134 | 56 | 0.185 | 0.190 | 103 | 1.125 | 0.648 | 58 | 829.836 | 546.192 | 66 |
| 0 g/L | 0.052 | 0.001 | 2 | 1.748 | 1.359 | 78 | 0.500 | 0.214 | 43 | 2.033 | 0.892 | 44 | 489.003 | 444.873 | 91 |
| | | | | | | **CA3-18** | | | | | | | | | |
| 1 g/L | 0.004 | 0.001 | 23 | 2.340 | 1.382 | 59 | 0.059 | 0.068 | 116 | 0.709 | 0.060 | 8 | 49.718 | 35.870 | 72 |
| 3 g/L | 0.093 | 0.001 | 1 | 2.315 | 1.184 | 51 | 0.081 | 0.050 | 62 | 0.549 | 0.036 | 7 | 47.229 | 24.658 | 52 |
| 7 g/L | 0.003 | 0.001 | 40 | 2.560 | 1.868 | 73 | 0.061 | 0.142 | 233 | 0.414 | 0.094 | 23 | 38.016 | 48.646 | 128 |
| 0 g/L | 0.003 | 0.001 | 33 | 2.724 | 1.438 | 53 | 0.077 | 0.098 | 127 | 0.484 | 0.055 | 11 | 46.985 | 39.998 | 85 |

(1) Regarding the reaction with nZVI, there is high similarity between zinc and cadmium (Figure 6); however, after 60 days, the average removal of Zn was higher than for Cd.

The removal of both elements in the interstitial water of the samples over the 120 days of testing can be summarized as follows:

(i)     After 60 days, the removal rate of Zn varied between 80% and 100%, and for Cd between 50% and 80%, with the last element corresponding to levels ranging from 0.6 mg/L to 3 mg/L. For Zn, for the higher nZVI concentrations, the average removal was approximately 99% (7 g/L) and 95% (3 g/L), with the corresponding concentrations ranging from 0.5 and 2.0 mg/L. Although with a significant reduction in relation to the initial contents, the highest removal rates of both elements from the soluble phase of the samples were not enough to keep their concentrations below the values tabulated for groundwater by the Legislation for the State of Minas Gerais, which is 1.05 mg/L for Zn and 0.005 mg/L for Cd [53].

(ii)    After 120 days, an increase of the levels of these two elements in the interstitial water of the samples was observed, particularly significant in CA2-10 for Zn (removal rates between 2% and 48%), which is the sample with the highest contamination index regarding these elements, the highest sulphate content, and the lowest permeability. However, when comparing the initial values before the injection of nZVI with the final values obtained in a composite sample, a marked decrease in both elements is perceptible in all the samples (Table 4).

(2)    For lead, the retention rate was directly proportional to the concentration of nZVI suspensions, and the suspensions with higher concentrations proved to be very efficient in its removal, in some samples, even after 120 days (Figure 7). This is the case of soil CA1-33, which, at the end of the test, showed a removal rate of 80–85%, corresponding to a concentration of 0.05–0.07 mg/L of Pb, albeit still higher than the regulated value of 0.01 mg/L [39]. Although with a maximum immobilization of 90% in the alluvium sample CA2-10, Pb decreased to about 0.08 mg/L. Only in sample CA3-18, values below the regulated value were reached, given the removal of almost 100% after 60 days, corresponding to a concentration of 0.001 mg/L of Pb. After this period, there was, however, a slight increase.

(3)    Copper is the element with the most distinct behavior regarding its retention by nZVI (Figure 7): (1) all the tests showed effective removal; and (2) contrary to the other elements, there is no proportionality between the retention rate and the suspensions concentration. In all the samples and for all the nZVI concentrations, the retention rate was always high, and the temporal evolution and behavior of Cu were very similar. After 60 days of the injection and at the final stage of the test (120 days), Cu concentrations in the interstitial solution were always lower than the regulated value of 2.00 mg/L [53]. The retention rates were very similar in the soil CA1-33 (70–95%) and in the alluvium CA2-10 (80–90%), reaching 99% in CA3-18. The minimum concentrations attained after 60 days corresponded to 0.02 and 0.04 mg/L, 0.1 and 0.3 mg/L, and 0.001 mg/L, respectively. In the case of CA3-18, given the high efficiency of the Cu removal by applying nZVI, the final values were very similar for any concentration of the suspension injected.

Analyses of the composite samples obtained by homogenization before the nZVI injection and after 120 days (Table 4) highlight the higher efficiency of the less concentrated suspensions in the Cu removal and the loss of reactivity of nZVI after 60 days of injection (Figure 7).

(4)    Although with critical values in the sedimentary materials and soil arsenic contents in the aqueous phase were very low, some of them slightly higher than the limits proposed by Brazilian legislation for groundwater [53].

In all the samples, As retention is quite irregular between different concentrations of the injected suspensions and between the top and bottom layers of each sample. In general, As removal was higher for the nZVI suspensions of 3 g/L and 7 g/L, and between 90 to

99% of this element was immobilized after 120 days (Figure 8). This immobilization rate corresponds to final concentrations of 0.001 mg/L, which is below the regulated value of 0.01 mg/L for groundwater [53]. The variation pattern of As removal from the soluble phase of the samples is different from that of the other elements, with a decrease in As levels in the first days after injection, an increase in the period between 14 and 60 days, and a further decrease at the final stage of the experimental study.

There seems to be a positive effect of As removal by nZVI, particularly for higher concentrations, 3 g/L and 7 g/L. The analyses of the composite samples obtained by homogenization at the beginning and at the end of the test highlight the effectiveness of As immobilization by the nanoparticles for any concentration (Figure 8).

However, it should be noted that the observed variations are not much representative, considering the low concentrations of this element in the aqueous phase of the samples, close to the legal limits for groundwater, and the insignificant improvement of the effect of nZVI injection on its removal when compared with water leaching (0 g/L, blank test).

### 4.3. Test Data Evaluation—Risk Spatial Projection

The data evaluation by risk spatial projection was carried out by comparing the levels of the target metals analyzed in the alluvial sediments and soil that were digested with *aqua regia*, before injection of the nZVI and at the final stage of the test, 120 days after injection.

In step one, the experimental variograms for each selected metal were calculated for the structural evaluation of the attribute. No clear evidence of anisotropies was found, and isotropic variograms were calculated and matching models were adjusted. The quality of the model of uncertainty provided by simple kriging (SK) (zero mean) was assessed using the same source and destination geography approach, whereby SK results at sampled locations were compared to observations. Correlation scores ranged from 0.70 to 0.85. Thus, the results of the cross-validation were found to be satisfactory for the selected models, indicating consistency between the estimated and observed values [1]. Furthermore, 100 simulations were performed using Sequential Gaussian Simulation (SGS) as a conditional stochastic simulation of the As, Cd, Cu, Pb, and Zn concentrations, on a 100 × 100 m grid to generate 100 equiprobable scenarios. Mean images (MI) were subsequently used to assess spatial hot-spots for contamination to each variable and therefore considered as risk maps (Figure 9). Spatial models are shown, and computed clusters allowed for the classification in growth risk zones [1]. The overlapping test samples (CA1-33, CA2-10, and CA3-18) for the efficacy assessment of nZVI indicate a substantial decrease in metal content and a substantial technique accuracy for metal retention.

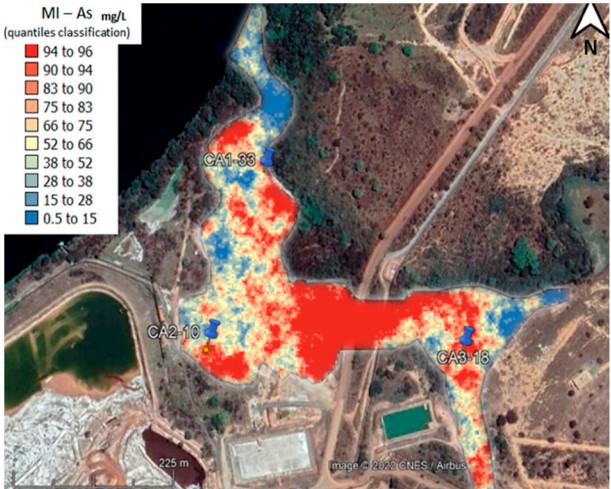
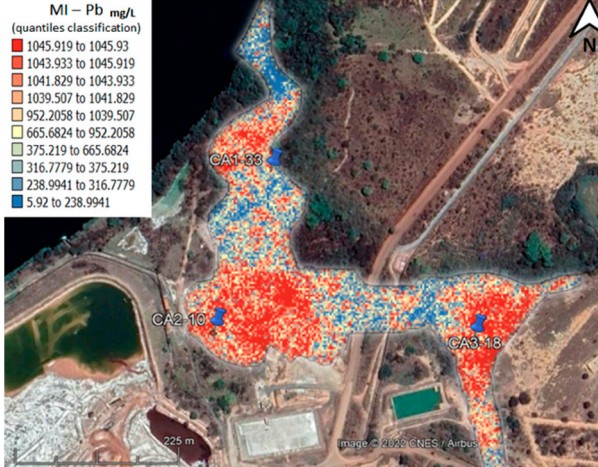

**Figure 9.** *Cont.*

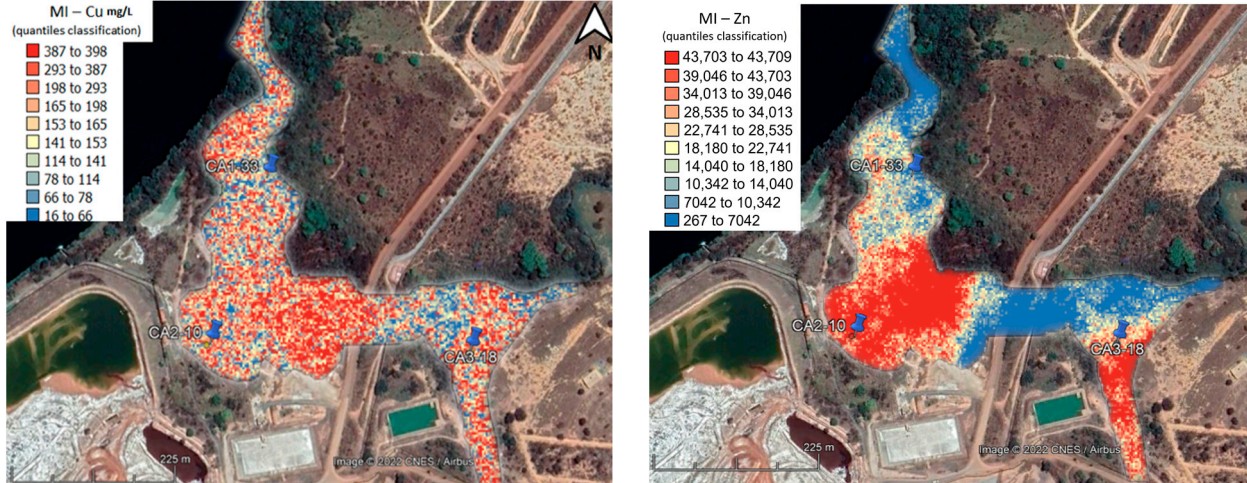

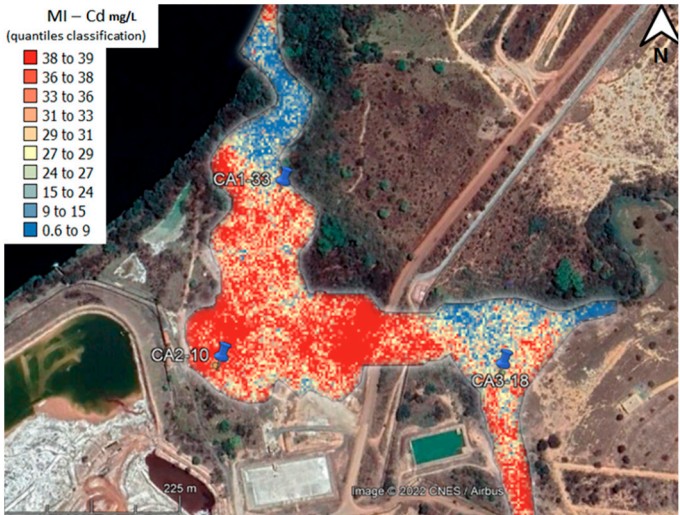

**Figure 9.** Sequential Gaussian Simulated mean images (MI) for As, Pb, Cu, Zn, and Cd. The test validation samples CA1-33, CA2-10, and CA3-18 were projected overlaying the corresponding simulated map. It is possible to acknowledge that the content level on the target validation points decreased considerably, therefore validating the zero-valent iron nanoparticles (nZVI) technology as a proxy for novel remediation approaches.

## 5. Discussion

### 5.1. Effect of a Laboratory-Scale Injection of nZVI Solutions into Heavily Metal-Contaminated Materials from a Tropical Climate

At the time of its implementation, this laboratory-scale nZVI injection was the first performed on highly contaminated sedimentary materials from a tropical climate, with very different characteristics from those in temperate climates. Thus, the behavior of the various metals after the injection of nZVI suspension may be different from those observed in previous works. An additional point to highlight is that in none of these studied cases [18–20,38,39,56–60], the concentrations of metals were so high and comprised such a distinct range of elements.

Since this is a laboratory-scale test, it cannot reflect the reality of what may happen at field scale, due to several factors that might have influenced the achieved results:

(1) In this experiment a drought situation was simulated, since the system was closed, and the extracted water was not replaced. The interstitial water became more concentrated over time and the material was always moist; however, there was no simulation of water loss by infiltration and water replacement by the rain effect. The most extreme

situation of a closed system was recreated, and thus, the concentrations of metals in soluble phase reached the highest values. If there is a marked decrease of the metal concentrations over time after nanoparticles application, this decrease should have greater expression at a real scale. Since not all the conditions of a real situation could be reproduced, the achieved data cannot be considered as absolute.

(2) The extractions of the interstitial water were always conducted in the same points of the sedimentary columns, in an upper and lower position; thus, the values obtained for the different metals do not reflect the whole column and they have to be considered as point sample values; this consideration results from the fact that aggregation and nanoparticles nuclei was verified in several areas of the columns and the diffusion of the particles through the soil and alluvial sediments was not uniform, due, mainly, to their low permeability. Models of aggregation of small particles have been published in many works and in most of them it is mentioned that a surface charge established on the surface of particles causes repulsive electrostatic forces between them. However, the iron particles corrode in the water, and this process can produce changes on the surface charge and on the aggregation rate [62]. Because particles are made from iron, they also have magnetic properties, which significantly affects the aggregation rate [28,63,64]. As a result of aggregate formation, the specific surface area will also decrease, resulting in a reactivity decrease [65].

(3) These two factors led to the development of distinct chemical environments in the same sample, which certainly conditioned different retention reactions and rates of the various metals. Thus, the irregularities verified for pH, redox, and element concentrations in the upper and lower layers of the columns corroborate the occurrence of different removal and solubilization reactions with distinct behaviors and intensities throughout the sample. The poor diffusion of the suspension may have been due, apart from the low permeability of the material, to the injection speed, which was possibly too high. However, since the injection was made in an upward direction, a lower speed could cause sedimentation of the particles in the peristaltic pump tubes.

(4) The data that will best reflect the real situation of metals immobilization correspond to the analysis of the composite porewater (Table 4) and levels of the target metals analyzed in digested samples, resulting from the homogenization of all the material in each column. However, these could only be carried out before the injection (t = 0) and in the final phase of the test (t = 120), when the soil and alluvium were removed from the columns after 4 months of the nanoparticles' injection.

*5.2. Factors Affecting the Variation of Physic-Chemical Parameters over the Batch Column Experiment*

The pH values strongly influence the redox reactions that occur on the surface of nZVI, either due to accelerated corrosion at low pH or by passivation of the nZVI surface through the formation of iron oxide hydroxides at high pH [56,57,66–68]. Passivation of the nZVI surface at high pH hinders electron transfer from the $Fe^0$ core, decreasing the removal of metal contaminants by reductive precipitation [56]. A high pH value also decreases the adsorption of metal anions due to the electrostatic repulsion caused by the negative charge of the nZVI surface. However, the negatively charged nZVI surface is favorable for the adsorption of metal cations, such as Cd, Zn, and Cu, with high contents in the study area, which are thus immobilized through precipitation by hydrolysis as metal hydroxides. In these metals, the pH effects are indicative of the existence of competition between protons and metal ions for the adsorption sites for specific pH values [56,57,66].

The increase of redox potential and decrease of pH of the interstitial water of samples after the first 2 days of nZVI injection, contrary to what was observed by most authors, e.g., [18–20,38,39,56–60], may possibly be explained by the very particular characteristics of these materials, very different from those tested in the abovementioned studies, namely:

(1) Chemical composition characterized by high levels of iron oxides ($Fe_2O_3$: 16–52%);

(2) High diversity of metals with extremely high contents under diverse chemical forms, mostly in easily soluble ones [1,7,9,11]. This diversity provides for chemical competition between the various metals for ionic adsorption sites on the nanoparticles [56].

(3) Excess of metals in solution in a confined environment, where no infiltration, diffusion, and leaching of soluble ions could occur, and where permanent water saturation might decrease oxidation conditions at an early stage. After 3 days, the ZVI might have reacted directly with the metallic elements in solution in cationic form, leading to $H^+$ release, with subsequent pH decrease and dissolution of some iron oxides, with high contents in these materials. This may also justify the increase of oxidation conditions, as can be demonstrated by Equation (2):

$$2Fe^0 + 3Zn^{2+} + 4H_2O \rightarrow 2FeOH + 3Zn^0 + 2H^+ \tag{2}$$

(4) In the presence of some metals, such as Pb, with high contents in these sedimentary materials, the reduction-oxidation reaction with the nZVI particles can lead to the formation of $H^+$, which may lower the pH (Equation (3)):

$$2Fe^0 + 3Pb^{2+} + 4H_2O \rightarrow 2FeOH + 3Pb^0 + 2H^+ \tag{3}$$

The heterogeneity and significant pH decrease of the water-saturated samples at the time zero of the test ($t_0$) in relation to the pH measured in situ in the sampling day ($t_{is}$), might be due to a change of the chemical conditions of the environment. Due to water saturation, hydrolysis reactions might have occurred, as the test was performed in a closed environment, which was conducive to the reaction of particles with water. In this reaction, there might have been the release of some cations responsible for the exchange acidity, namely $H^+$ and $Al^{3+}$.

The oscillations of Eh throughout the test were equivalent to those verified for pH, which corroborates the high interdependence between both parameters [56,67,68]. In each sample, the fluctuations of pH and Eh values observed in the four replicates (corresponding to three different concentrations of nZVI and the blank),which might be a consequence of different chemical conditions inside the column, namely different chemical reactions with the metal cations, which may have affected their solubility.

The conductivity measures the concentration of all the ionic elements in solution, including anionic complexes (e.g., sulphates, nitrates), alkaline, alkaline-earth, and metallic cations (contaminant metals and Fe). Thus, it is to be expected the sharp increase that was observed in this parameter right after the first day of nZVI injection, particularly in the samples corresponding to the injection of the most concentrated suspension (7 g/L).

When $Fe^0$ of the nucleus of the nZVI particles yields electrons, it turns to $Fe^{2+}$ and $Fe^{3+}$, which may pass into solution, remaining in soluble phase for low pH values [14,16,56]. This pH condition was observed in the first days of the test (with pH values < 3 in some samples after 3 days). The gradual increase of pH from 7 days of injection onwards might contribute to a reduction of the mobility of Fe, which could create suitable conditions for the precipitation of iron oxides, thus decreasing their concentration in the soluble phase. The immobilization of iron is followed by other metallic cations that were in high concentrations in the aqueous phase, such as Zn and Cd. These elements can co-precipitate with iron or be adsorbed on the iron hydroxides ($FeOOH$, $Fe(OH)_3$) formed on the surface of the nanoparticles, which have a strong retention capacity [13,14,16–19,56]. These retention mechanisms occurred approximately after 7 days and may explain the gradual decrease of conductivity values from that day on.

### 5.3. Removal Mechanisms of Contaminants by nZVI

The specific removal mechanisms associated to the treatment of heavy metal contamination with ZVIs depend on the standard redox potential ($E^0$) of the metal contaminant [41].

1. Metals that have an $E^0$ that is more negative than, or similar to, that of $Fe^0$ (e.g., Cd and Zn) are removed from solution by adsorption onto the iron oxide (hydroxide) layer surrounding the zero-valence iron ($Fe^0$) core. Upon binding to the FeOOH layer, these metals bond through electrostatic interactions without undergoing changes in their valence state. To a lesser extent, complexation at the surface of the nanoparticles and co-precipitation may also occur.

2. Metals with an $E^0$ much more positive than $Fe^0$ (e.g., As and Cu) are preferentially removed by precipitation and layer-mediated reduction (reductive precipitation) on the surface of nZVI [45].

3. Metals with $E^0$ slightly more positive than $Fe^0$ (e.g., Pb) can be removed by both adsorption and partial chemical reduction.

Oxidation and co-precipitation by iron oxides are the other possible reaction mechanisms, depending upon the prevailing geochemical conditions (pH, Eh) and the initial concentration and speciation of the contaminant metals.

The metal-nZVI interactions for the target metals included in this study can, therefore, be categorized as [41]:

1. Reduction—As, Cu, and Pb.
2. Adsorption—As, Pb, Cd, and Zn.
3. Oxidation/reoxidation—As and Pb.
4. Co-precipitation—As.
5. Precipitation—Cu, Pb, Cd, and Zn.

*Immobilization of zinc and cadmium*

There was a similar variation in the retention rates and conditions of both elements, which may be due to several aspects: (1) occurring in highly mobile chemical forms, (2) increased solubility by decreasing pH, and (3) identical retention mechanism by nZVI.

The variation of the retention rates of Zn and Cd in function of the pH oscillations denotes the strong influence of this parameter on the solubility of both elements.

The effect of pH variations may also explain the decrease of Zn and Cd in the samples after the injection of water (0 g/L). The chemical reactions were triggered by water, among which hydrolysis may cause oscillations in the pH values, influencing the higher or lower retention/mobility of these two elements. In these sedimentary materials, mostly composed by minerals like quartz, Fe-oxides (magnetite, often oxidized to hematite or goethite, feldspars, and micas (muscovite and partly chloritized Fe-biotite) [9], hydrolysis caused by injected water may contribute to pH oscillations, as we have two very distinct mineralogical assemblages: (1) the silicates that behave like a salt of a strong base and a weak acid, and (2) the Fe-oxides that function like a salt of a weak base and a strong acid [69]. In these reactions, water spontaneously ionizes into hydronium cations ($H_3O^+$) and hydroxide anions ($OH^-$). While in the first case the cations tend to remain in ionic form and react very little with the $OH^-$ ions, contributing to an increase in pH, in the second case, the anion will become a spectator ion and fail to attract the $H^+$, while the cation from the weak base will donate a proton to the water forming a hydronium ion ($H_3O^+$) [69]. This last reaction will decrease the pH of the environment.

The injection of the material with nZVI can be considered an efficient process in reducing the solubility of Cd and Zn, especially up to 2 months after the injection of the suspension. The data obtained indicate the suspension with the highest concentration (7 g/L) as the most effective in removing both. Although with high efficiency in the immobilization of the two elements, their final concentrations never reached the values regulated for groundwater [54], possibly due to several factors that may have compromised a higher retention capacity by nZVI:

(1) Extremely high contents, much higher than the toxicity limits.

(2) The high levels of these two elements in soluble forms may influence the retention capacity of nZVI, through competition between the two cations for chemo-adsorption sites in the (oxy)iron hydroxide layer formed on the nanoparticles surface. It has been

observed by several authors [17] that, in the presence of these two cations and in view of the competitiveness between both, there is a more efficient removal and selectivity of nZVI particles for $Zn^{2+}$ than for $Cd^{2+}$.

(3) Low pH values that decreased after the first day of the test and remained low until about 60 days after injection. Low pH accelerates the corrosion and the dissolution of the oxide layer of the nZVI, increasing the reaction rates due to greater availability of electrons from the $Fe^0$ core [56]. Therefore, the general decrease of pH after 2 days of injection may have contributed to intense reactions that may have rapidly immobilized metallic cations by adsorption onto the iron oxide (hydroxide) layer surrounding the zero-valence iron ($Fe^0$) core. However, the impact of pH on metal removal by nZVI depends on the oxidation state of the metal and the removal mechanism [56]. For Zn and Cd, besides the very high values in solution, both cations are easily mobilized at pH < 5.5 [70]. Thus, although a very significant immobilization of most metals was observed, the concentrations of Zn and Cd, with retention rates ranging from 80–100% for Zn and 50–80% for Cd, were not able to reach the legislated values. The high reaction rates due to the low pH values are likely to have been one of the factors responsible for the decreased reactivity of nZVI after 2 months.

(4) Presence of high levels of sulphates, which is also an inhibiting factor for Zn and Cd retention, since it is a competitive anion for receiving electrons transferred by the nZVI [56]. Sulphates, in contact with nZVI, may be reduced to sulphides, which precipitate with metal cations on the surface of nanoparticles, reducing their adsorption capacity.

*Immobilization of Lead*

Through the retention mechanism called adsorption with partial chemical reduction, lead can be removed from the aqueous phase by nZVI via reduction to $Pb^0$ and by adsorption in the $Pb^{2+}$ form. In this reaction, $Pb^{2+}$ ions precipitate as $Pb(OH)_2$ and also undergo oxidation as $\alpha$-$PbO_2$ [56]. Considering the Pb concentrations after 120 days of the injection, all the samples showed a very sharp decrease in relation to the initial phase (t = 0), which demonstrates the high efficiency of nZVI in the immobilization of this element. The chemical reduction seems to be a very important mechanism in the decrease of the soluble contents of Pb, which can be supported by the significant decrease of its concentration only by water saturation, which corresponds to the blank test.

*Immobilization of Copper*

The behavior of copper concerning the reactions with the nanoparticles is different from the other analyzed metals because it is an element with standard redox potential much more positive than $Fe^0$, having therefore distinct removal mechanisms. Through the preferential mechanism of chemical reduction, $Cu^{2+}$ can be removed in its elemental form or be reduced to $Cu^+$ resulting in the formation of $Cu_2O$ that cements to the surface of the nanoparticles [56].

The application of nZVI was effective in removing most of the Cu in solution, and the tests were more efficient when using less concentrated suspensions. The decrease of Cu solubility only by water saturation (blank test) reveals the effectiveness of washing the soil and sediments keeping saturation conditions for the immobilization of this element. Under saturation, the oxygen limitation may have been a sufficient condition for the Cu retention by reduction, followed by precipitation. According to several authors [71], Cu retention is independent of pH values, being more dependent on the oxygen content of the environment. In fact, among the analyzed elements, Cu was the one that showed the least dependence on the pH decrease.

*Immobilization of Arsenic*

Although it occurs as an anionic complex, arsenic behaves similarly to copper when reacting with zero-valent iron nanoparticles, because both have a similar standard redox potential ($E^0$) that is much more positive than $Fe^0$, and therefore have identical reactions with the core of the particles and with the hydroxide (oxide) layer around them. According

to several authors [14,16,19,39] on this element, nZVI has high functionality: (1) As may complex with the iron oxide layer and can be removed from solution by co-precipitation with iron sulphides, oxides, and hydroxides; (2) As can be reduced to $As^{3+}$ or $As^0$; (3) the $As^{3+}$ ions thus formed and $As^{5+}$ ions (the most common ionic form) can be adsorbed or co-precipitated on the nZVI surface; and (4) some $As^{3+}$ ions can be oxidized to $As^{5+}$, either by action of $OH^-$ groups or by action of iron oxides, both formed by oxidation of zero-valence iron ($Fe^0$), thereby forming iron oxide–$As^{3+}$ complexes.

In this experiment, since the diffusion of the nZVI suspensions throughout the samples was not homogeneous, and particle aggregates were formed in several areas, and different chemical environments may have been created within each column, which could have resulted in different reactions of the nanoparticles. However, the most likely explanation is related to the very low levels of this element in the soluble phase of the sediments and soil, though the very high contents in the extractable fraction, so that the observed oscillations are of little importance.

### 5.4. Aging Time of nZVI

For most of the metallic elements, a very similar evolution was observed, a small decrease of their solubility after the injection of nZVI during the first 24 or 48 h, followed by an increase of their contents in the mobile phase until about 30 days, indicating the formation of new, more soluble species, mostly due to the decrease of the pH values registered during the same period. From 30 days and up to 60 days after injection, a very significant decrease in the concentration of most metals was verified, some of them to levels below the intervention limits. During this period, in which there was a pH increase, adsorption, precipitation, complexation, or co-precipitation reactions should have occurred, leading to the retention of a large proportion of metals by the nanoparticles.

In general, a decrease of the removal capacity of these elements was observed after 60 days of injection, suggesting the loss of reactivity of the nanoparticles.

The decrease in reactivity, which follows the decrease in pH values, may result from the inhibition of the formation of iron oxide and hydroxide on the nZVI surface due to the release of $H^+$, resulting in less adsorption of metal cations. Besides the lowering of the pH of the system, the high lithogenic iron contents naturally occurring as crystalline and amorphous iron oxides, the very high levels of metallic elements in easily soluble forms and the reactions that may occur in the $Fe^0/H_2O$ system may have contributed to the oxidation of the injected nZVI approximately 2 months after their injection, causing the loss of their reactivity in a period much shorter than expected. Furthermore, the particles may have aggregated, and a new layer of mixed valence oxide ($Fe^{2+}$-$Fe^{3+}$) may have been formed [72,73]. Oxidation/aging of the nZVI results in the loss of $Fe^0$ content, and therefore less reductive $Fe^0$ becomes available to react with the target contaminant [74,75].

After 60 days of injection, the retained species may have been desorbed, moving back to the soluble phase, a situation that was observed after 120 days. This finding agrees with some works [75,76] showing that a second injection of nanoparticles is usually necessary after 60 days, due to the loss of reactivity, which leads to desorption processes and competition between chemical species.

### 5.5. Reduction of the Risk Level of Sediments and Soils after nZVI Injection

The overlapping of the risk spatial projection of the test samples, based on the extractable contents of the target metals (obtain by *aqua regia* digestion) before the injection of the nZVI and at the final stage of the test, after 120 days, indicated a significant decrease of the risk level. Although the immobilization of these metals from the porewater was more effective 60 days after the injection, after which a further increase in their soluble levels was observed, data from digested sediments and soil indicate the effectiveness of this nanotechnology in reducing the levels of contaminating metals, not only in soluble forms, but also in extractable non-soluble forms.

A possible explanation for the better results obtained by the sedimentary materials regarding the decrease of extractable forms (soluble and non-soluble) of metals at the end of the test may be related to reactions between ZVI nanoparticles and some mineral components, where metals are preferentially associated. These reactions may transform them into more stable crystalline forms, not extractable by digestion with *aqua regia*. According to some authors [60,77], in the nZVI reaction, metallic iron is oxidized in the presence of water, removing other metal ions from aqueous media by chemical absorption. In this process, crystalline iron oxides, such as hematite ($\alpha$-$Fe_2O_3$), goethite ($\alpha$-$FeO(OH)$), and magnetite ($Fe^{2+}Fe^{3+}_2O_4$), can be produced, leading to the immobilization of other metals, which were initially in soluble forms or loosely bound to other inorganic compounds (e.g., Mn oxides, amorphous Fe oxides, and exchangeable cations). These Fe-oxides are among the most stable and sparingly soluble in acid media in the visible range and are most reactive in a UV range or in the presence of reductant and/or complexing agents [77]. Thus, by digestion of the sediments and soil with aqua regia, they may not have been completely solubilized.

These reactions may be more extensive in Fe-rich materials, as is the case of the tested materials.

## 6. Conclusions

In addition to solving a serious environmental problem in the surrounding area of a large industrial plant in Southeast Brazil, this study aimed to fill the gap that exists in the application of nZVI in the decontamination of fine-grained alluvial sediments and soils, with high proportions of a wide range of metals in a tropical region dominated by Fe-rich lithologies.

Even if many studies have demonstrated the success of nZVI in situ remediation in the immobilization of a large range of heavy metals, there are still many uncertainties that must be addressed, such as the possible formation of microsized cluster due to the nanoparticle aggregation and the mobility lack of bare nZVI. These aggregations also appeared in this column experiment at a laboratory level, creating different chemical conditions in the same sample and sometimes hindering the interpretation of the data obtained. The rapidly nanoparticles aggregation and the settling out of the aqueous suspensions posed one significant obstacle, limiting a wider application of nZVI over the whole length of the samples to be decontaminated.

In addition to the phenomena of natural aggregation of particles, in this study there are also a few other factors that it is important to highlight to improve the effectiveness of the nZVI technology as a strategy to reduce the potential leachability of metals in similar contaminated scenarios to prevent their transport into deeper soil layers, rivers, and groundwater:

I.   This was a laboratory batch study where it was not possible to simulate the drainage, diffusion, and precipitation conditions that naturally occur. This corresponded to a very complex and closed system, tested under extreme conditions, simulating a prolonged period of drought, in which there was no replacement of water during the entire test; thus, the interstitial water of the materials became progressively more concentrated.

II.  This experiment reflected the real situation of the surrounding area of a metallurgical plant, where most of the soils, alluvium, and river sediments showed very high concentrations of heavy metals and sulphates of anthropic origin, including high levels of lithogenic iron and manganese. The contaminant metals include elements with different standard redox potentials ($E^0$) relative to that of $Fe^0$, which led to multiple retention mechanisms, namely, adsorption, desorption, reduction, oxidation, complexation, and co-precipitation. This high diversity under soluble phase decreased the reactivity of the nZVI particles, probably by competition among the various cations for the exchange sites or, in the case of anionic complexes such as sulphates, by reduction and transformation into sulphides that may be precipitated together with metallic cations on the surface of the nanoparticles.

Despite all these limitations, data from the analysis of selected heavy metals (Zn, Cu, Cd, and Pb) and As in the extractable and in the dissolved phases of this highly contaminated sediments/soils indicate the nZVI injection as a suitable technique for the reduction of the risk level of PTEs in contaminated Fe-rich tropical environments. The nZVI particles were effective in reducing metals mobility in the porewater, although concentrations below the limit regulated by Brazilian legislation for groundwater were rarely reached, and the nanoparticles lost reactivity before the period that would be expected, 60 days after injection. The comparison of the extractable contents of these target metals (obtain by *aqua regia* digestion) before the injection of the nZVI and at the final stage of the test after 120 days also indicated the efficiency of this nanotechnology in reducing the levels of pollutant metals, not only in soluble forms, but also in non-soluble forms.

The environmental remediation by nZVI technology has been widely tested in column experiments but, at a laboratory level, results may differ significantly and may not reflect full-scale conditions, so it is critical to conduct a pilot test in the field before proceeding with full-scale remediation. In general, the results of a pilot test will provide a more representative view of reality and will be more suitable for a full-scale design. The pilot test will also provide important information on the practical conditions for implementation under site-specific circumstances, such as (i) the amount of suspension that should be injected and the injection pressure, (ii) the radius of influence of the injection and therefore the required distance between the injection points, (iii) prediction of the pollutant load reduction and the possible after-treatment values to be achieved, (iv) the effects on the geochemistry of soils or sedimentary materials, and (v) the timing of the nZVI reactivity.

In light of a few limitations and of the insufficient understanding of some of the derived hazards, six points could be considered in future studies in order to reduce these limitations: (1) Avoid the lowering of pH and, consequently, the increase of metal solubility (especially for Zn and Cd) by using a buffer solution simultaneously with the injection of nanoP. (2) Consider the loss of reactivity of nZVI after 60 days and make a second injection after this period, in order to renew reactivity, thus keeping the metals immobilized. After this period, a control should be made to ensure that the contaminating elements remain immobilized. (3) Given the high concentrations of many metallic elements in soluble forms and the loss of reactivity of the nanoparticles after 60 days, the use of macro-scale zero-valent iron particles is thought to be more effective. Nanoparticles promote a faster removal of contaminants; however, they have a shorter life span, which, in the case of highly contaminated environments with metals with distinct chemical behaviors, could make their use economically unviable. (4) The low permeability of these soils and sediments makes the particles injection not the best treatment process. With materials of this nature and with such high levels of contaminants, a more effective technique could be, possibly, the use of microscale ZVI (mZVI), as a filling of reactive barriers, through which the water contained in the materials will flow. (5) Since the use of ZVI particles in contaminated materials is not a technique to reduce the levels of metals but rather to immobilize them, the possibility of their desorption and recovery could be evaluated. This study could contribute to the strategic concept of circular economy, which is based on the reduction, reuse, recovery, and recycling of materials and energy. (6) Evaluation of the effectiveness of bimetallic iron-based nanoparticles (BNPs) in these highly contaminated sediments/soils because, according to some authors [14], the presence of two different metals in the BNPs induces a synergic effect that gives more favorable properties to improve the degradation of such a complex mixture of metals.

**Author Contributions:** Conceptualization, R.F.; methodology, R.F., C.P., T.A. and J.A.; software, T.A. and J.A.; validation, R.F., T.A. and J.A.; formal analysis, R.F. and C.P.; investigation, R.F., resources, R.F.; data curation, R.F. and T.A.; writing—original draft preparation, R.F. and J.A.; writing—R.F., T.A. and J.A.; visualization, R.F. and T.A.; supervision, R.F.; project administration, R.F.; funding acquisition, R.F. All authors have read and agreed to the published version of the manuscript.

**Funding:** This research was funded by Votorantim Metais S.A. Company through the consultant Project "Proposal of remediation strategy of Consciência and Barreiro Grande streams-phase 2". Some of the equipment used in this study was purchased under the project INALENTEJO—Quadro de Referência Estratégia Nacional 2007–2013 (QREN) through the projects ALENT-07-0262-FEDER-001867 and ALENT-07-0262-FEDER-001876.

**Acknowledgments:** The authors acknowledge the funding provided by ICT, under contract with FCT (Portuguese Science and Technology Foundation) under the Project FCT—UIDB/04674/2020. The authors are also grateful for the helpful comments and suggestions of the two Reviewers.

**Conflicts of Interest:** The authors declare no conflict of interest.

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
