# Peer review of "Evaluating the Effectiveness of Nanotechnology in Environmental Remediation of a Highly Metal-Contaminated Area—Minas Gerais, Brazil"

_geosciences, doi:10.3390/geosciences12080287_

Round 1
Reviewer 1 Report
Comments are attached

Author Response
Évora, July 13th 2022
Dear Reviewers
We are grateful for your positive appraisal of our manuscript and for your comments, which have been fundamental to the improvement of our manuscript “Evaluating the effectiveness of nanotechnology in environmental remediation of a high metal-contaminated area - Minas Gerais, Brazil”.
Please find below our replies to those comments. Whenever we do not agree with the suggestions or comments, we provide a suitable rebuttal.
We hope we have satisfactorily addressed all the relevant questions and improved our manuscript accordingly.

Reviewer 2 Report
This work aims to test the effectiveness of an in situ remediation methodology based on the immobilization of these PTEs through the application of zero-valent iron nanoparticles (nZVI). It contains some new results which may be interesting to some Geosciences readers. Therefore, I recommend publication of the manuscript after addressing the following comments:
The manuscript must be edited by a senior author or an English professional to improve English level along with some minor corrections. Some of the English errors are listed below:
“…were cross-checked with the…” should be replaced with “…was cross-checked with the…”.
“…suitable technique for reduction the risk…” should be replaced with “…suitable technique for reducing the risk…”.
“…once nano dimensional and stable at superficial conditions.” should be replaced with “…once nano-dimensional and stable under superficial conditions”.
“behaviour" should be replaced with “behavior” all over the manuscript.
“…part of the of Três Marias formation…”
“…it was placed washed and disinfected…”
“…the nZVI are an important reducing agent that, in aqueous media, react…”.
The abstract needs improvement. Please add more information on the research results. It should be informative and completely self-explanatory and should not include experimental procedure.
The introduction should start with stating the objectives of the work and providing an adequate background, avoiding a summary of the results.
Figures should be labeled as (a), (b), (c)… followed by a descriptive caption or title.
Figures captions should clearly describe the figures with more detail.
Following references should be cited to improve the quality of the paper:
International Journal of Hydrogen Energy 2019, 44 (44), 24162-24173. DOI: https://doi.org/10.1016/j.ijhydene.2019.07.129.
Industrial & Engineering Chemistry Research 2020, 59 (1), 183-193. DOI: 10.1021/acs.iecr.9b05130.
Journal of Cleaner Production 2020, 275, 124157. DOI: https://doi.org/10.1016/j.jclepro.2020.124157.
Ultrasonics Sonochemistry 2020, 64, 105044. DOI: https://doi.org/10.1016/j.ultsonch.2020.105044.
The results and discussion section should be improved. It should be presented with clarity and precision, should be explained by referring to the literature, and should interpret the findings in view of the obtained results.
Author Response

(The authors gave the same response as above.)

Round 2
Reviewer 2 Report
The authors have addressed all of my comments and concerns in the revised version. Overall, the manuscript is well written and the reported results are of valuable interests to readers. I have no additional comments and I recommend accepting the paper.